# Fundamental limits and non-reciprocal approaches in non-Hermitian quantum sensing

Hoi-Kwan Lau[1] & Aashish A. Clerk[1]

Unconventional properties of non-Hermitian systems, such as the existence of exceptional points, have recently been suggested as a resource for sensing. The impact of noise and utility in quantum regimes however remains unclear. In this work, we analyze the parametric-sensing properties of linear coupled-mode systems that are described by effective non-Hermitian Hamiltonians. Our analysis fully accounts for noise effects in both classical and quantum regimes, and also fully treats a realistic and optimal measurement protocol based on coherent driving and homodyne detection. Focusing on two-mode devices, we derive fundamental bounds on the signal power and signal-to-noise ratio for any such sensor. We use these to demonstrate that enhanced signal power requires gain, but not necessarily any proximity to an exceptional point. Further, when noise is included, we show that non-reciprocity is a powerful resource for sensing: it allows one to exceed the fundamental bounds constraining any conventional, reciprocal sensor.

---

[1] Institute for Molecular Engineering, University of Chicago, 5640 South Ellis Avenue, Chicago, IL 60637, USA. Correspondence and requests for materials should be addressed to H.-K.L. (email: hklau.physics@gmail.com)

Among the most powerful and ubiquitous measurement techniques is dispersive measurement, where a parameter of interest shifts the frequency of a resonant electromagnetic mode. Dispersive measurement is used in a myriad of tasks, including in settings where quantum noise and quantum limits are relevant. Examples range from the sensing of biomolecules and nanoparticles[1–3], to the measurement of superconducting qubits[4,5], quantum optomechanical measurements of mechanical motion[6], and gravitational wave detection[7–9].

Given its widespread utility, methods for improving dispersive measurements are of immense practical and fundamental interest. In this regard, there has been considerable recent interest in exploiting non-Hermitian dynamics in linear coupled-mode systems to enhance dispersive-style measurements[10–16]. Such systems are described by an effective non-Hermitian Hamiltonian matrix, and can exhibit exceptional points (EPs), where as a function of parameters two eigenvalues of the Hamiltonian coalesce and the matrix becomes defective. Near such EPs, the system eigenvalues have an extremely strong dependence on small changes in parameters. In the simplest two-mode realization[17,18], a parameter $\epsilon$, which enters the Hamiltonian linearly is able to shift eigenmode frequencies by an amount $\sqrt{\epsilon}$. For small $\epsilon$, this suggests an extremely strong response, and the possibility of enhanced sensing. The first experiments probing this extreme sensitivity of mode frequencies to parametric changes have recently been reported[19,20].

To date, almost all work on EP-based sensing implicitly assumes frequency shifts whose magnitude is at least comparable to mode linewidths. It is, however, also interesting to ask whether non-Hermitian sensing methods are effective in the common weak dispersive regime, where frequency shifts are small; this is the goal of our work. Analyzing this regime involves addressing several general questions about non-Hermitian sensing. First, most studies focus exclusively on characterizing parametric shifts of mode frequencies; the process of how such shifts are measured is not fully analyzed. This is problematic, as a realistic sensing protocol may be sensitive to the parametric dependence of both the eigenvalues and eigenvectors of the system Hamiltonian; this latter dependence could conceivably counteract the parameter dependence of the eigenvalues[21]. Second, the impact of fluctuations has not been discussed. In the coupled-mode settings of interest, non-Hermitian effective dynamics always corresponds to dissipative dynamics which will generically be accompanied by noise. This noise can limit the ability to resolve parameter changes. This is especially crucial in quantum settings, where one can never ignore the effects of vacuum noise, especially if the dissipative dynamics involves amplification processes.

In this paper, we address both these sets of issues. We analyze a generic linear non-Hermitian sensing setup by mapping it to a probability-conserving open quantum system (see Fig. 1a). This allows us to fully account for fluctuation effects in both classical and quantum regimes of operation. We analyze an optimal homodyne-based measurement scheme, and derive fundamental bounds on signal power and signal-to-noise ratio (SNR) that constrain sensing protocols based on non-Hermitian Hamiltonians. Focusing on two-mode sensors, we find that the apparent improvement of signal power in a system with an EP can be accomplished by simply adding gain to a more conventional setup without any EP. Further, when fluctuation effects are included, we find that the SNR of any reciprocal system (whether or not it exploits an EP) is constrained by a fundamental bound involving the average intramode photon number. Surprisingly,

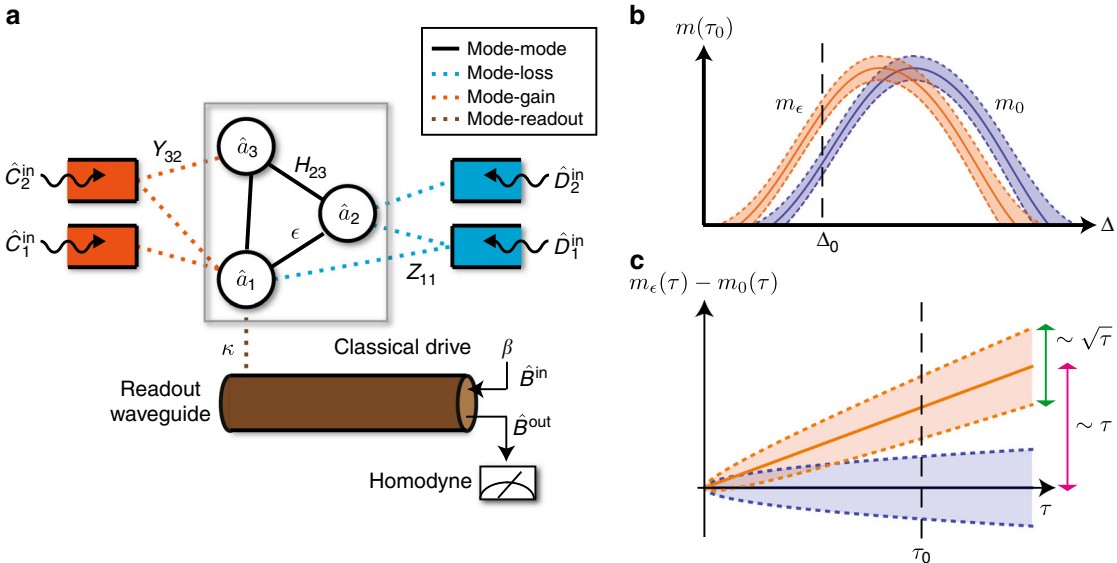

**Fig. 1** General dispersive measurement setup and measurement results. **a** The setup consists of resonant modes (circles) that interact via a parameter-dependent non-Hermitian Hamiltonian. Standard analyses only consider the non-Hermitian dynamics of mode amplitude (region inside gray rectangle). In this work, we instead treat the system as an open quantum system, where non-Hermitian dynamics is generated by coupling to gain/loss baths (red/blue rectangles) and a readout waveguide. The operators and parameters are defined in Eq. (5). Particularly, the coupling rates to the various baths (dotted lines) is characterized by matrices $Y$ and $Z$, and $H$ describes the Hermitian direct couplings between modes (solid lines). A classical drive is injected into the readout waveguide, which couples only to mode 1. Its reflected field is measured by homodyne detection. A parametric change in the Hamiltonian (e.g., coupling between modes 1 and 2 here) changes the state of modes as well as that of the reflected field. **b** Integrated homodyne current $m$ at a certain measurement time $\tau_0$, as a function of the detuning $\Delta$ of the drive frequency from the cavity 1 resonance frequency. The shaded area denotes uncertainty due to measurement noise, and the two curves are for two values of the parameter to be sensed. A parametric change can be optimally detected by measuring at a single detuning, e.g., $\Delta_0$ (dashed vertical line). **c** Time variation of integrated homodyne current for a fixed detuning $\Delta_0$. The signal induced by the perturbation to be sensed ($\sqrt{\mathcal{S}}$, pink arrow) scales linearly as $\tau$, while the uncertainty ($\sqrt{\mathcal{N}}$, green arrow) has a weaker scaling, $\sqrt{\tau}$. Therefore, any small perturbation can be resolved for sufficiently long $\tau$

sensors involving nonreciprocal interactions can surpass this bound; we analyze this in detail. We thus identify nonreciprocity, which is accessible in a variety of experimental platforms, as a novel resource for enhanced sensing.

## Results

**General setup**. We consider a generalized version of the non-Hermitian sensing system studied in previous works[10,13–15,19,20,22]: $M$ resonant modes interact as described by the linear and Markovian coupled-mode equations:

$$\dot{\alpha}_i(t) = -i \sum_j \tilde{H}_{ij}[\epsilon] \alpha_j(t). \tag{1}$$

$\alpha_j(t)$ denotes the amplitude of mode $j$, and the $M \times M$ matrix $\tilde{H}$ is an effective non-Hermitian Hamiltonian describing both coherent and dissipative linear dynamics. The Hamiltonian depends on a parameter $\epsilon$, and the goal is to sense an infinitesimal change in $\epsilon$. We assume that this parameter only changes non-dissipative terms in $\tilde{H}$, and thus write

$$\tilde{H}_{ij}[\epsilon] = \tilde{H}_{ij}[0] + \epsilon V_{ij} \tag{2}$$

where the Hermitian matrix $V$ describes the coupling of the parameter to the dynamics. We take $\epsilon$ to have units of frequency, and hence $V$ is dimensionless.

Unlike many works, we explicitly analyze the protocol used to measure the parametric dependence of $\tilde{H}$. A general strategy is to couple mode 1 to an input–output waveguide or transmission line, and then use this port to drive this system with a coherent tone at a frequency $\omega_{dr}$. The reflected signal is then measured, and used to infer $\epsilon$. Coupling to the waveguide introduces extra damping of mode 1, and hence $\tilde{H}_{ij} \to \tilde{H}_{ij} - i(\kappa/2)\delta_{i1}\delta_{j1}$, where $\kappa$ is the coupling rate to the waveguide. Working in a rotating frame at the drive frequency, the coupled-mode equations now become:

$$\dot{\alpha}_i = i\Delta \alpha_i - i \sum_j \tilde{H}_{ij}[\epsilon] \alpha_j - i\delta_{i1}\sqrt{\kappa}\beta, \tag{3}$$

where $\beta$ is the amplitude of the coherent drive. Without loss of generality, we take $\beta$ to be real and positive, and choose a frequency reference such that $\operatorname{Re} \tilde{H}_{11}[0] = 0$. This implies that $\Delta$ represents the detuning of the drive frequency from the mode-1 resonance frequency.

In addition to fully treating the measurement, we also want to consistently describe noise effects associated with the dissipative dynamics encoded in $\tilde{H}$. Dissipative dynamics correspond to the anti-Hermitian part of $\tilde{H}$, which can always be written in terms of the difference of two positive-definite matrices. We thus write

$$\frac{1}{2i}\left(\tilde{H} - \tilde{H}^\dagger\right) \equiv YY^\dagger - ZZ^\dagger - \frac{1}{2}\tilde{\kappa}, \tag{4}$$

where $\tilde{\kappa}_{ij} = \kappa \delta_{i1}\delta_{j1}$. The matrix $YY^\dagger$ represents gain processes, i.e., processes that tend to cause exponential growth in time; correspondingly, $ZZ^\dagger$ represents loss processes (beyond the loss associated with the input–output waveguide). For definiteness, we take $Y$ to be a $M \times N_Y$ matrix, and $Z$ to be a $M \times N_Z$ matrix. We also define $H = \left(\tilde{H} + \tilde{H}^\dagger\right)/2$ (i.e., the Hermitian part of $\tilde{H}$, which describes nondissipative dynamics).

We can now view the coupled-mode equation in Eq. (3) as the noise-averaged version of a fully probability-conserving linear Markovian open quantum system. This description is useful even in the classical regime if one wants to account for the effect of thermal noise. The non-Hermitian dynamics in $\tilde{H}$ are generated by coupling to $N_Y + N_Z$ distinct dissipative environments, with specific mode-bath coupling constants given by the

matrices $Y$, $Z$. Letting $\hat{a}_i$ denote the canonical bosonic annihilation operator of the $i$th mode, the full system is described by the Heisenberg–Langevin equations:

$$\dot{\hat{a}}_i = i\Delta \hat{a}_i - i\sum_j \tilde{H}_{ij}[\epsilon]\hat{a}_j - i\delta_{i1}\sqrt{\kappa}\beta$$
$$\qquad - i\delta_{i1}\sqrt{\kappa}\hat{B}^{in} - i\sqrt{2}\left(\sum_{j=1}^{N_Y} Y_{ij}\hat{C}_j^{in\dagger} + \sum_{j=1}^{N_Z} Z_{ij}\hat{D}_j^{in}\right) \tag{5}$$

The first line here has the same structure as in Eq. (3), and describes the linear dynamics of our system and its coherent driving. The terms on the second line instead describe zero-mean noise driving our system. $\hat{B}^{in}$ is noise entering from the input–output waveguide, whereas $\hat{C}_j^{in}\left(\hat{D}_j^{in}\right)$ are noises entering from the dissipative baths used to realize the gain (loss) parts of the dissipative dynamics encoded in $\tilde{H}$. Consistent with the linear, Markovian nature of our system, these noise operators represent (operator-valued) Gaussian white noise. Quantum mechanically, they cannot be zero: at best, they describe vacuum fluctuations. In this case, we have:

$$\left\langle \hat{Q}^{in}(t)\hat{Q}^{in\dagger}(t')\right\rangle = \left(\bar{n}_Q^{th} + 1\right)\delta(t - t') \tag{6}$$

$$\left\langle \hat{Q}^{in\dagger}(t)\hat{Q}^{in}(t')\right\rangle = \bar{n}_Q^{th}\delta(t - t') \tag{7}$$

$$\left\langle \hat{Q}^{in}(t)\hat{Q}^{in}(t')\right\rangle = 0 \tag{8}$$

where $Q \in \{B, C_j, Z_j\}$, and there are no correlations between different noise operators. The averages above represent averages over different realizations of the noise process, or equivalently, over the state of the bath degrees of freedom. $\bar{n}_Q^{th}$ represents the thermal occupancy of bath $Q$; we focus on the case where there is only vacuum noise, and these occupancies vanish (though our formalism can also easily treat the classical case $\bar{n}_Q^{th} \gg 1$).

Note that Eq. (5) describes the same average dynamics as our starting coupled-mode equations: taking the average of Eq. (5) and defining $\alpha_i \equiv \langle \hat{a}_i \rangle$ recovers Eq. (3). The additional noise effects encoded in Eq. (5) will, however, be important in determining our ability to make a measurement. We stress that these Markovian Heisenberg–Langevin equations are standard in the study of open quantum systems; a derivation is provided in Methods, and pedagogical treatments are given in[5,23].

A crucial observation here is that the system-bath coupling matrices $Y$, $Z$ in Eq. (4) are not uniquely determined by $\tilde{H}$. This ambiguity corresponds to a simple physical fact: there are many different ways to couple to dissipative baths to realize a given non-Hermitian dynamics. As is perhaps obvious, noise will play a crucial role in determining the measurement sensitivity of $\epsilon$; hence, the sensitivity will depend on the particular choice of baths and bath couplings used to realize $\tilde{H}$. This leads to two important conclusions: first, $\tilde{H}$ on its own does not completely specify the performance of our detector, and second, for a given non-Hermitian Hamiltonian, an optimal measurement will require using an optimized choice of dissipative baths and bath couplings.

**Homodyne measurement and measurement rate**. We now discuss how the information on $\epsilon$ in the reflected field leaving mode 1 can be extracted. We will characterize the measurement sensitivity using standard metrics that are well established in describing a weak, continuous linear measurement; see, e.g.[5], for a

pedagogical discussion. This will allow us to directly compare the non-Hermitian sensing protocols to more established methods.

The amplitude of the reflected field in the waveguide is described by an operator $\hat{B}^{\text{out}}$. Using standard input–output theory[23], we have

$$\hat{B}^{\text{out}}(t) = \left(\beta + \hat{B}^{\text{in}}(t)\right) - i\sqrt{\kappa}\hat{a}_1(t). \qquad (9)$$

The first term describes the incident field on mode 1 that is promptly reflected, whereas the second term describes the field emitted from mode 1. Note that the reflected field in our geometry is completely equivalent to the transmitted field in standard setups where an optical fiber is coupled to a whispering-gallery mode resonator[1–3].

For small $\epsilon$, the average value of the output field will have a linear dependence on $\epsilon$. We will be interested throughout this paper on long measurement times, and hence will focus on the steady state (time-independent) value of this average. We thus write

$$\langle\hat{B}^{\text{out}}\rangle_\epsilon \simeq \langle\hat{B}^{\text{out}}\rangle_0 + \lambda\epsilon \qquad (10)$$

where $\lambda$ is a (possibly complex) linear response coefficient. We throughout use $\langle..\rangle_z$ to denote an average calculated using Eq. (5) with $\epsilon = z$.

Letting $\phi = -\arg\lambda$, it is clear that all the information on $\epsilon$ in the output field is contained in the real part of $e^{i\phi}\hat{B}^{\text{out}}$. An optimal measurement strategy is thus to measure this quantity directly. This corresponds to one quadrature of the output field, and the necessary measurement is known as homodyne detection. The time-dependent measurement signal (i.e., the homodyne current) is described by the operator $\hat{I}(t)$:

$$\hat{I}(t) \equiv \sqrt{\frac{\kappa}{2}}\left(e^{i\phi}\hat{B}^{\text{out}}(t) + e^{-i\phi}\hat{B}^{\text{out}\dagger}(t)\right) \qquad (11)$$

Note the factor of $\sqrt{\kappa}$ is included in the homodyne current for convenience, as it makes $\hat{I}$ have the units of a rate.

The homodyne current will be subject to shot noise fluctuations which will obscure our ability to extract $\epsilon$. This noise is described by a spectral density[5]:

$$\bar{S}_{II}[\omega] = \frac{1}{2}\int_{-\infty}^{\infty} dt\, e^{i\omega t}\left\langle\left\{\delta\hat{I}(t), \delta\hat{I}(0)\right\}\right\rangle_0 \qquad (12)$$

where $\delta\hat{I} \equiv \hat{I} - \langle\hat{I}\rangle$. As we are considering the effects of an infinitesimal perturbation $\epsilon$, we can characterize our measurement sensitivity using the noise spectral density calculated to zeroth order in $\epsilon$.

To estimate $\epsilon$, the homodyne current is integrated from $t = 0$ to $t = \tau$ to average away the effects of noise. The time-integrated measurement is thus described by the operator:

$$\hat{m}(\tau) \equiv \int_0^\tau dt\,\hat{I}(t). \qquad (13)$$

Considering the long-$\tau$ limit, the power associated with the signal induced by the perturbation is:

$$\mathcal{S} = \left[\langle\hat{m}(\tau)\rangle_\epsilon - \langle\hat{m}(\tau)\rangle_0\right]^2 = 2\kappa\epsilon^2|\lambda|^2\tau^2 \qquad (14)$$

We have assumed a measurement time $\tau$ that is long enough that we can ignore any transient effects in the behavior of $\langle\hat{I}(t)\rangle$. Note also that with our definitions, $\mathcal{S}$ is dimensionless.

Similarly, the noise power associated with the integrated homodyne current in the long-time limit is:

$$\mathcal{N} \equiv \langle\delta\hat{m}(\tau)\delta\hat{m}(\tau)\rangle_0 = \tau\bar{S}_{II}[0], \qquad (15)$$

where $\delta\hat{m} \equiv \hat{m} - \langle\hat{m}\rangle_0$.

Combing these results, we see that the SNR of signal power associated with the homodyne measurement grows linearly with time:

$$\frac{\mathcal{S}}{\mathcal{N}} = 2\kappa\epsilon^2\tau\frac{|\lambda|^2}{S_{II}[0]} \equiv \frac{\epsilon^2}{\kappa^2}\tau\Gamma_{\text{meas}}. \qquad (16)$$

We have defined the long-time linear growth of the SNR in terms of a measurement rate $\Gamma_{\text{meas}}$. This is a standard metric for quantifying the resolving power of weak continuous measurements; $(\kappa/\epsilon_0)^2\Gamma_{\text{meas}}^{-1}$ represents the minimum time required to distinguish $\epsilon = \epsilon_0$ from $\epsilon = 0$. The measurement rate defined here is also directly related to the another standard metric for sensitivity, the imprecision noise spectral density[5].

More fundamentally, one could ask whether homodyne measurement is truly the optimal way to use the output field to estimate $\epsilon$. While heuristically this seems clear from Eq. (10), one can ask the question more formally. The maximum amount of information available in the output field considering all possible measurements is quantified by the quantum Fisher information[24]. This quantity can be calculted exactly for our linear, Gaussian system[25]. In Methods, we show that this metric coincides with the SNR given above in the limit where the driving field $\beta$ is sufficiently large. As such, the homodyne measurement strategy here is indeed the optimal strategy.

We stress that our measurement scheme involves driving the system at a single frequency only. This is in contrast to most works on EP sensing[10,13,19,20], which involve probing the system over a wide range of frequencies to measure a full output field spectrum. This approach requires driving the system with multiple tones, with each drive tone contributing to the total photon number (circulating power) in the coupled modes. To make a meaningful comparison between different schemes, we imagine a situation where the total photon number (in all modes and over all frequencies) is constrained. The question is then whether in this setting, it is better to have a multitone drive, or a drive at just a single frequency. We find that there is no advantage for such multitone driving, as the information generated at each frequency is independent. It is thus optimal to probe the system with a single coherent tone whose frequency is chosen to optimize $\Gamma_{\text{meas}}$, see Fig. 1. We provide a rigorous proof of this statement (in terms of the quantum Fisher information) in Methods.

**General expressions and constraints for a linear system.** While the definition of the SNR and measurement rate in Eq. (16) is generally applicable, things simplify enormously for our system given the linearity of the dynamics. For a stable system, the Langevin equations in Eq. (5) can be solved in the Fourier domain in terms of the dimensionless system susceptibility matrix $\tilde{\chi}$ defined as

$$\tilde{\chi}[\omega; \Delta; \epsilon] \equiv i\kappa\left[(\omega + \Delta)\mathbb{I} - \tilde{H}[\epsilon]\right]^{-1}, \qquad (17)$$

where $\mathbb{I}$ is the $M \times M$ identity matrix. Using the input–output relation in Eq. (9) and taking average values, we immediately find that the steady-state average homodyne current is given by

$$\langle\hat{I}\rangle = \sqrt{2\kappa}\,\text{Re}\left[e^{i\phi}\beta\left(1 - \tilde{\chi}_{11}[0; \Delta; \epsilon]\right)\right] \qquad (18)$$

Note that the homodyne current depends on $\epsilon$ through $\tilde{\chi}$, which in turn depends on both the eigenvalues and eigenvectors

of $\tilde{H}$. The zero-frequency susceptibility matrix can in general be written in terms of the eigenvalues $\Omega_j$ of $\tilde{H}$ as

$$\tilde{\chi}[0; \Delta; \epsilon] = -i\kappa \frac{\text{adj}(-\Delta\mathbb{I} + \tilde{H}[\epsilon])}{\prod_j(-\Delta + \Omega_j[\epsilon])}, \quad (19)$$

where $\text{adj}(\cdot)$ is the adjugate matrix. The basis of many sensing techniques is that the eigenvalues $\Omega_j$ generally have a dependence on $\epsilon$, which directly influences the susceptibility and hence output field. However, to get a complete description of the measurement, one must also worry about the numerator in this expression: the adjugate matrix (e.g., right and left eigenvectors of $\tilde{H}$) will also in general depend on $\epsilon$, which can serve to suppress the overall sensitivity to $\epsilon$. In what follows, we thus focus on the entire susceptibility matrix, and not just on the eigenfrequencies of $\tilde{H}$.

Returning to Eq. (18) and considering small $\epsilon$, one readily finds a direct expression for the linear response coefficient $\lambda$ in Eq. (10). Defining $\chi(\Delta) \equiv \tilde{\chi}[0; \Delta; 0]$ as the zero-frequency, unperturbed susceptibility matrix, we have

$$\lambda = -\beta \frac{d\tilde{\chi}_{11}[0; \Delta; \epsilon]}{d\epsilon}\bigg|_{\epsilon=0} = i\frac{\beta}{\kappa}(\chi V \chi)_{11}. \quad (20)$$

We will implicitly assume $\chi$ is evaluated at $\Delta$ unless specified.

Using this expression, it is straightforward to calculate the signal power associated with the time-integrated homodyne current (c.f. Eq. (14)):

$$\frac{\mathcal{S}}{(\epsilon\tau)^2} = 2\frac{\beta^2}{\kappa}|(\chi V \chi)_{11}|^2 = 2\bar{n}_{\text{tot}}\frac{|(\chi V \chi)_{11}|^2}{(\chi^\dagger \chi)_{11}}. \quad (21)$$

In the second equality, we have expressed $\mathcal{S}$ in terms of the total average photon number in all modes induced by the coherent drive:

$$\bar{n}_{\text{tot}} \equiv \sum_i \left\langle \hat{a}_i^\dagger \right\rangle \langle \hat{a}_i \rangle = \frac{\beta^2}{\kappa}\sum_i |\chi_{i1}|^2 = \frac{\beta^2}{\kappa}(\chi^\dagger \chi)_{11}. \quad (22)$$

Our motivation here is that $\mathcal{S}$ can always be increased indefinitely by simply increasing the drive power. For a meaningful metric, one thus needs to ask how much signal is generated given a fixed number of photons used for the measurement. In many situations, the photons to worry about are the intracavity photons described by $\bar{n}_{\text{tot}}$: if this photon number becomes too large, a variety of problems typically ensue (e.g., unwanted heating effects and breakdown of linearity). Note that we have neglected the incoherent photons injected by the gain bath. Unlike the drive-induced photon number that scales linearly as the driving power, the incoherent photon number is independent of coherent drive. For a sufficiently large driving power, the drive-induced photon thus dominates the total photon number.

Turning to the fluctuations in the homodyne current, a straightforward calculation using Eqs. (5) and (15) yields:

$$\bar{S}_{II}[0] = \frac{\kappa}{2}\left(1 + \frac{4}{\kappa}(\chi Y Y^\dagger \chi^\dagger)_{11}\right). \quad (23)$$

The first term here represents the unavoidable shot noise in the homodyne current. The second term describes additional noise emanating from the dissipative baths that generate the gain processes in $\tilde{H}$. This extra noise corresponds to the amplification of zero-point fluctuations, and is connected to the fact that quantum mechanically, phase-insensitive linear amplification cannot be noiseless[26]. We stress that for a fixed $\tilde{H}[0]$, the choice of $Y$ is not unique; thus, the noise properties of our setup is not

directly determined by $\tilde{H}[0]$, but will depend crucially on how the dissipative dynamics is realized using external baths.

For a fixed $\tilde{H}[0]$ (and hence fixed $\chi$), we can find the optimal choice of baths and bath couplings that minimizes the noise in the homodyne current (see Methods). We find:

$$\bar{S}_{II}[0]_{\text{min}} = \frac{\kappa}{2}\left(1 + 2\Theta\left[|1 - \chi_{11}|^2 - 1\right]\left(|1 - \chi_{11}|^2 - 1\right)\right), \quad (24)$$

where $\Theta[z]$ is the Heaviside step function. Again, this result reflects the well known quantum limits on added noise of linear amplifiers[26]. Here, if our system has reflection gain (i.e., $|1 - \chi_{11}| > 1$), the output noise must be bigger than simple shot noise. We stress that for any given $\tilde{H}[0]$ and corresponding susceptibility matrix $\chi$, it is always possible to construct a realization of the dissipative dynamics (in terms of bath couplings $Y$, $Z$) that attains this minimum possible noise level (see Methods).

Combining these results gives us a general bound on the measurement rate of any linear system:

$$\Gamma_{\text{meas}} \leq \Gamma_{\text{opt}} \equiv \frac{4\kappa\bar{n}_{\text{tot}}}{(\chi^\dagger \chi)_{11}}\frac{|(\chi V \chi)_{11}|^2}{1 + 2\Theta\left[|1 - \chi_{11}|^2 - 1\right]\left(|1 - \chi_{11}|^2 - 1\right)}. \quad (25)$$

This fundamental bound on the measurement rate (i.e., long-time SNR) of our linear sensor will now allow us to compare the best possible performance of sensors with different Hamiltonians. More precisely, we want to quantitatively ask whether systems exploiting non-Hermitian physics (such as EP-based sensing schemes) can ever offer advantages over more conventional sensing schemes, including simple sensing schemes based on a linear amplifier.

Note that while our focus is on sensing a parameter which modifies the Hermitian part of the Hamiltonian, Eqs. (21) and (25) can also be used directly in the more general case where $V$ is non-Hermitian. The only additional assumption required is extremely mild: we just need that any extra noise associated with the non-Hermitian parameter change is not pathologically large, i.e., it vanishes in the limit $\epsilon \to 0$, where the parametric dependence of the Hamiltonian vanishes. As long as this is satisfied, the extra noise would not change the form of the bound on the SNR in Eq. (25).

**Two-mode non-Hermitian sensors for coupling perturbation.** The results above are extremely general, applying to any non-Hermitian sensing setup described by Eq. (5). In Methods, we consider the simple case where the parameter of interest simply shifts the resonant frequency of mode 1. Here, however, we apply our results to the specific kind of system that has been extensively studied in the literature on EP sensing[10,13–15,19,20]: a two-mode system described by a non-Hermitian Hamiltonian $\tilde{H}[\epsilon]$, where the parameter to be determined is a Hermitian coupling between the modes. This corresponds to a coupling matrix

$$V = \begin{pmatrix} 0 & 1/2 \\ 1/2 & 0 \end{pmatrix} \quad (26)$$

in Eq. (2).

The signal power in the homodyne current follows directly from Eq. (21) and is given by

$$\mathcal{S} = \frac{1}{16}\frac{|\chi_{11}|^2|\chi_{12} + \chi_{21}|^2}{|\chi_{11}|^2 + |\chi_{21}|^2}\mathcal{S}_\epsilon. \quad (27)$$

where

$$\mathcal{S}_\epsilon \equiv 8\epsilon^2 \tau^2 \bar{n}_{\text{tot}}. \tag{28}$$

is the signal power associated with a standard, single-mode dispersive measurement (see Methods).

**Reciprocal two-mode sensors.** Consider first a reciprocal system, where the magnitude of the coupling between the two modes does not have any directionality, i.e., $|\tilde{H}_{12}| = |\tilde{H}_{21}|$. Note that this definition of reciprocity here is consistent with the standard usage of a scattering matrix being invariant under exchange of source and receiver[27]. If we attached a weak coupling waveguide to mode 2, the condition $|\tilde{H}_{12}| = |\tilde{H}_{21}|$ would ensure that the amplitudes for $1 \to 2$ transmission and $2 \to 1$ transmission have the same magnitude.

This immediately implies that $|\chi_{12}| = |\chi_{21}|$, and allows us to bound the maximum value of $\mathcal{S}$:

$$\mathcal{S}_{\text{recip}} \leq \frac{1}{4} \mathcal{S}_\epsilon |\chi_{11}|^2. \tag{29}$$

Thus, for a reciprocal system, the only way to parametrically increase the signal power (at fixed measurement time $\tau$ and intracavity photon number $\bar{n}_{\text{tot}}$) is to make $|\chi_{11}|$ large. This implies that the system is an amplifier: signals incident in the coupling waveguide will be reflected with gain.

Including now the effects of measurement noise, the above bound on signal power for a reciprocal two-mode system, when combined with Eq. (46), immediately yields a bound on the measurement rate:

$$\Gamma_{\text{meas,recip}} \leq 16\kappa \bar{n}_{\text{tot}}. \tag{30}$$

We see that $\Gamma_{\text{meas}}$ for a reciprocal sensor is fundamentally bounded by the intracavity photon number and the coupling rate $\kappa$ to the waveguide; unlike signal power, it cannot be made

arbitrarily large by increasing $|\chi_{11}|$. As discussed in Methods, achieving this bound requires $\chi_{11} = 2$, implying the absence of reflection gain. If one instead increases $|\chi_{11}| \gg 1$ to achieve a large signal power, the optimal measurement rate instead approaches $2\kappa \bar{n}_{\text{tot}}$.

These results apply directly to the kind of non-Hermitian two-mode sensors that have been studied extensively in the literature[14–16,19]. These systems generically involve a sensing parameter that couples as per Eq. (26), and a reciprocal two-mode effective Hamiltonian of the form

$$\tilde{H}_{\text{recip}}[0] = \begin{pmatrix} -i\frac{\kappa+\gamma_1}{2} & J \\ J & -i\frac{\gamma_2}{2} \end{pmatrix}. \tag{31}$$

Here, $J$ is the Hermitian coupling between the modes, whereas $\gamma_1$, $\gamma_2$ describe possible gain/loss processes (depending on the sign) acting locally on each mode. As always, $\kappa$ represents the coupling rate between the input–output waveguide and mode 1; note that this coupling has mostly been neglected in previous work.

The eigenvalues of $\tilde{H}[0]$ in this case are:

$$\Omega_\pm[0] = -i\frac{\kappa+\gamma_1+\gamma_2}{2} \pm \sqrt{J^2 - \frac{1}{4}(\kappa+\gamma_1-\gamma_2)^2}. \tag{32}$$

It thus exhibits a stable EP when $J = (\kappa + \gamma_1 - \gamma_2)/4$ and $\kappa + \gamma_1 + \gamma_2 > 0$. For this tuning of $J$, the mode eigenvalues behave as $\Omega_\pm[\epsilon] = \pm\sqrt{2J\epsilon} - i(\kappa + \gamma_1 + \gamma_2)/2$, and have a strong square-root dependence on $\epsilon$.

Despite the large sensitivity of mode frequencies to $\epsilon$ at the EP, the signal power and measurement rate for this setup remain bounded by Eqs. (29) and (30). This is shown explicitly in Fig. 2, where the signal power and measurement rate for this system is plotted as a function of the drive frequency. These quantities never exceed the fundamental bounds.

Note that in many applications, it is only the signal power that is relevant, as the measurement noise will be limited by non-

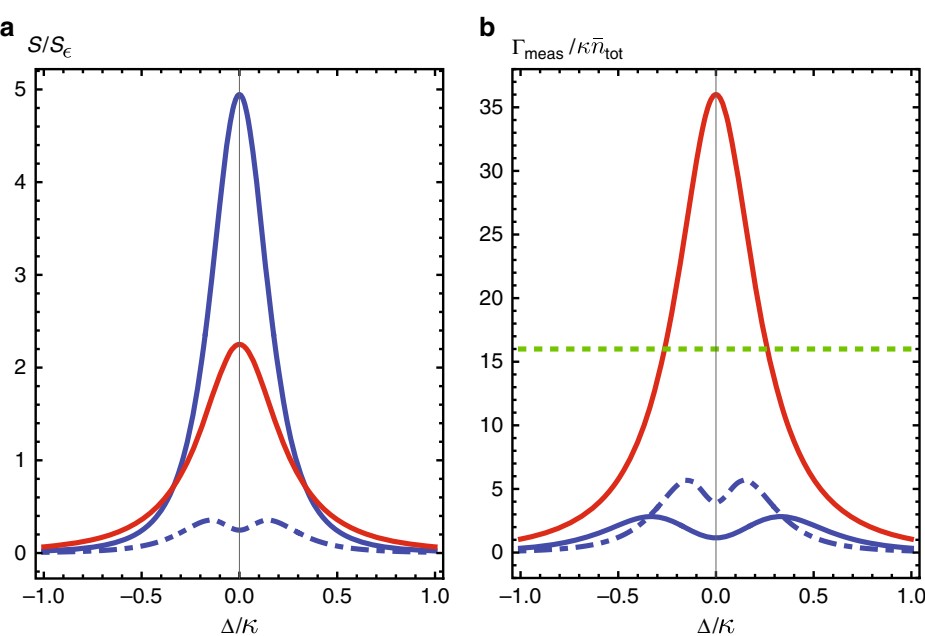

**Fig. 2** Signal power and measurement rate. **a** Signal power $\mathcal{S}$ and **b** measurement rate $\Gamma_{\text{meas}}$ against drive detuning $\Delta$ for three 2-mode non-Hermitian sensors. Blue dot-dashed: Reciprocal system with exceptional point but no gain, described by Eq. (31) with $\gamma_1 = 0$, $\gamma_2 = 0.2\kappa$, $J = 0.2\kappa$. Blue solid: Reciprocal system with exceptional point and gain, described by Eq. (31) with $\gamma_1 = 0$, $\gamma_2 = -0.3\kappa$, $J = 0.325\kappa$. Despite a higher signal power, introducing gain does not enhance the measurement rate due to the corresponding increased level of measurement noise. Neither of these systems beat the fundamental reciprocal-system bound in Eq. (30) (green dotted). Red: nonreciprocal system in Eq. (35) ($\gamma_1 = \kappa$, $\gamma_2 = 0.5\kappa$, $J = 1.5\kappa$, $\nu_2 = 0$). It yields a measurement rate which appreciably exceeds the reciprocal-system bound for a wide range of $\Delta$

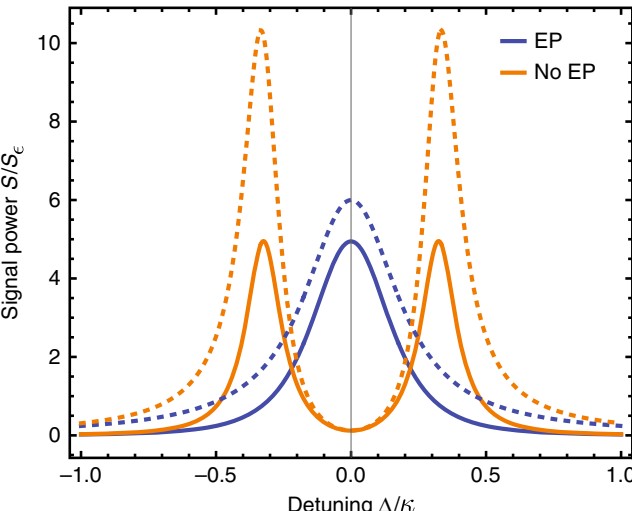

**Fig. 3** Signal power versus drive detuning for two reciprocal two-mode sensors. An EP system (blue, described by Eq. (31) with $\kappa + \gamma_1 = \kappa$, $\gamma_2 = -0.3\kappa$, $J = 0.325\kappa$), and a simple two-mode amplifier system that never has an EP (orange, Eq. (31) with $\kappa + \gamma_1 = \gamma_2 = 0.16\kappa$, $J = 0.325\kappa$). The two systems have similar peak signal powers. Dotted lines denote bound on signal power for both systems as given by Eq. (29)

intrinsic effects (e.g., following amplifiers and detector inefficiency). It is thus interesting to note that the signal-power performance of the two-mode EP system in Eq. (31) can be matched with a simple two-mode amplifier setup, where the first mode is subject locally to gain, and the total damping rate of mode 2 is made to match that of mode 1. While this system never possesses an EP, its performance matches the EP system, see Fig. 3. Thus, in terms of signal power at fixed photon number, there is no fundamental utility here to using a EP system.

**Nonreciprocal two-mode sensors**. The above discussion shows that for a reciprocal system, tuning to an EP does not provide special advantages for measurement. We now consider another means of exploiting non-Hermitian physics: a sensor whose effective Hamiltonian breaks reciprocity, i.e., $\left|\tilde{H}_{12}\right| \neq \left|\tilde{H}_{21}\right|$. Breaking reciprocity allows one to parametrically exceed the bounds in Eqs. (29) and (30) that constrain any reciprocal two-mode sensing system. Synthetic nonreciprocity in driven photonic systems is an active area of current research (see ref. 28 and references therein), with experimental demosntrations in photonic platforms as well as superconducting quantum circuits and optomechanical systems. While most work in this area has focused on achieving nonreciprocal scattering to build devices such as isolators and circulators, we show here that nonreciprocity can also be a powerful resource for enhanced sensing.

To see how nonreciprocity changes our sensing problem, consider again Eq. (27) for the signal power, in the extreme directional limit where $\chi_{21} = 0$, but $\chi_{12} \neq 0$. This describes a situation where mode 2 influences mode 1 but not vice-versa. The signal power for this fully directional setup becomes independent of $\chi_{11}$:

$$\mathcal{S}_{\text{dir}} = \frac{1}{16}\mathcal{S}_\epsilon \left|\chi_{12}\right|^2. \tag{33}$$

The signal power could now in principle be increased indefinitely by increasing $\chi_{12}$ while keeping the intracavity photon number and reflection gain fixed. We illustrate the intuition behind this effect in Fig. 4.

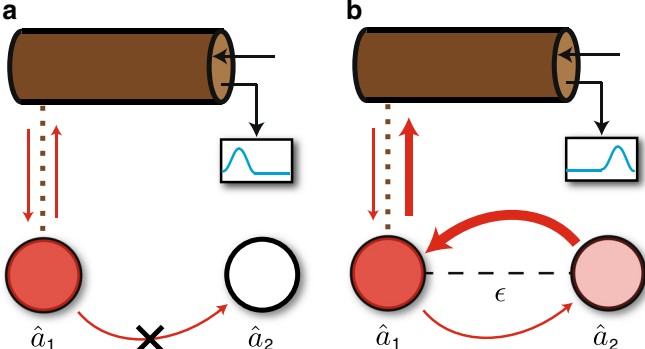

**Fig. 4** Schematic showing how signal power is enhanced by nonreciprocity. Arrows and their thicknesses illustrate the direction and magnitude of influences. **a** In the absence of any perturbation, nonreciprocity prevents mode 2 being excited by the drive incident on mode 1. Drive photons do not experience the large nonreciprocal tunneling $J$ from mode 2 to 1. **b** A nonzero coupling perturbation ($\epsilon \neq 0$) breaks nonreciprocity. As a result, drive photons can tunnel from 1 to 2, and then return experiencing amplification from the large nonreciprocal tunneling amplitude $J$. This gives a large signal

The benefits of nonreciprocity are more apparent when we consider noise and the full expression for the measurement rate. In a nonreciprocal system we can increase the signal power indefinitely (by making $\chi_{12}$ large) without having to have a large $\chi_{11}$ and hence reflection gain. This implies that the output noise can stay at the shot noise level. For a nonreciprocal system, we thus have:

$$\Gamma_{\text{meas,dir}} \leq \kappa \bar{n}_{\text{tot}} \left|\chi_{12}\right|^2 \tag{34}$$

For $\left|\chi_{12}\right| > 4$, this exceeds the fundamental bound on the measurement rate of a reciprocal system given in Eq. (30). We thus see that nonreciprocity is a resource for enhanced sensing; moreover, it does not require a system that is tuned to an EP.

It is helpful to consider a concrete example of a fully nonreciprocal setup. Consider a non-Hermitian Hamiltonian

$$\tilde{H}_{\text{dir}}[0] = \begin{pmatrix} -i\frac{\kappa + \gamma_1}{2} & J \\ 0 & \nu_2 - i\frac{\gamma_2}{2} \end{pmatrix}, \tag{35}$$

where $\gamma_1$ and $\gamma_2$ describe local damping or antidamping of the two modes, $\nu_2$ is the frequency detuning of the two modes, and $J$ describes a (complex) nonreciprocal mode-mode coupling. Such directional couplings can be realized in many different ways. For example, one could start with a three-mode system in a ring geometry with purely Hermitian couplings that have nontrivial complex phases. As discussed extensively in ref. 29, if one then adds strong damping to the third mode and adiabatically eliminates it, one can realize an effective two-mode Hamiltonian identical to that in Eq. (35).

The susceptiblity matrix is readily found. One has $\chi_{21} = 0$. For a drive that is resonant with mode 1 (i.e., $\Delta = 0$), the remaining elements are

$$\chi_{11} = \frac{2\kappa}{\kappa + \gamma_1}, \quad \chi_{12} = -\chi_{11}\frac{J}{\nu_2 - i\gamma_2/2}. \tag{36}$$

As desired, one can make $\chi_{12}$ arbitrarily large by increasing $J$ without requiring that $\chi_{11}$ also become large. As a result, one can reach the upper bound on the measurement rate given in Eq. (34) for $\gamma_1 \geq 0$. The performance of this nonreciprocal sensor is shown in Fig. 2, where its performance is compared against reciprocal

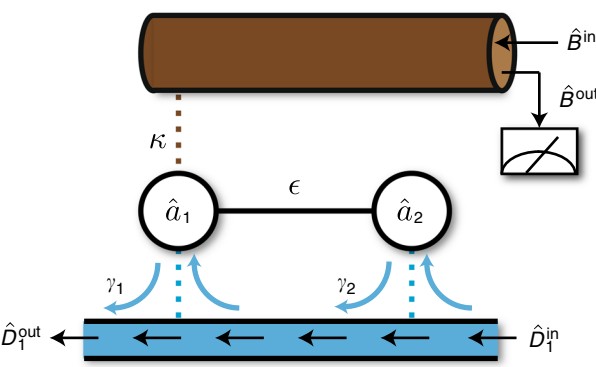

**Fig. 5** Implementation of a simple nonreciprocal two-mode sensor. Both modes are coupled to a single effective chiral waveguide; no coupling to gain baths is required. This system is capable of arbitrarily exceeding the fundamental bound on the measurement rate of any reciprocal two-mode sensor. The required chiral waveguide could be realized using circulators, dynamic modulation[28], or by using driven parametric interactions and external dissipation[29]. Symbols follow that in Fig. 1 and $\gamma_i \equiv |Z_{i1}|^2$

non-Hermitian systems. One clearly sees the violation of the reciprocal-system bound on the measurement rate.

Note that at an EP, the Jordan normal form of a $2 \times 2$ matrix has the nonreciprocal form of Eq. (35), but is also constrained to have identical diagonal entires; this was pointed out in ref. 13. We stress however that the benefits of our nonreciprocal setup have nothing to do with tuning our system to an EP or having eigenvalues coalesce. To see this explicitly, note that the unperturbed eigenvalues of $\tilde{H}_{\text{dir}}[0]$ are

$$\Omega_-[0] = -i\frac{\kappa + \gamma_1}{2}, \quad \Omega_+[0] = \nu_2 - i\frac{\gamma_2}{2}. \quad (37)$$

The system has an EP only when the parameters are precisely tuned to $\nu_2 = 0$ and $\kappa_1 + \gamma_1 = \gamma_2$. In contrast, the large enhancement of the measurement rate we obtain only requires $|J| \gg \sqrt{\nu_2^2 + \gamma_2^2/4}$. This condition is clearly unrelated to the presence of an EP.

While the simple nonreciprocal sensing setup in Eq. (35) is capable of reaching the fundamental bound in Eq. (34), whether or not this occurs depends on exactly how the dissipative dynamics encoded in $\tilde{H}_{\text{dir}}$ is realized through couplings to external baths. Here, it is possible to achieve the needed non-Hermitian Hamiltonian using only passive dissipation (i.e., no coupling to gain baths, $\mathbf{Y} = 0$ in Eq. (4)). The simplest realization would involve coupling both modes to an effective chiral waveguide, as depicted in Fig. 5; the (positive) coupling rate between mode $j$ and the waveguide is denoted $\gamma_j$. Focusing on the case of two modes with identical frequencies (i.e., $\nu_2 = 0$), and using standard cascaded quantum systems theory[23] to describe this setup, we realize the non-Hermitian Hamiltonian in Eq. (35) with $J = -i\sqrt{\gamma_1\gamma_2}$. Further, as there are no couplings to gain baths, the homodyne current noise is always given by its minimal shot noise value. This setup then realizes the optimal value for the measurement rate for a nonreciprocal setup as given in Eq. (34). Setting $\gamma_1 = \kappa$, we have:

$$\Gamma_{\text{meas}} = 4\kappa\bar{n}_{\text{tot}}\left(\frac{\gamma_1}{\gamma_2}\right) \quad (38)$$

Comparing against Eq. (30), we see that this system beats the reciprocal-system measurement rate bound whenever $\gamma_2 < (1/4)\gamma_1$.

We note that there are a variety ways of implementing a coupling to an effective chiral waveguide. These range from

conventional approaches based on the use of circulators, to realizations of chiral waveguides using topological photonic systems[30], to methods that mimic chiral propagation by using dynamic modulation and engineered dissipation[28,29,31,32]. We stress that such engineered nonreciprocal interactions have been experimentally realized in photonic setups[33–35], classical microwave circuits[36,37], optomechanical systems[38,39], and superconducting circuits[40,41]. While the motivation for these experiments was largely to build circulators and isolators, our work shows that such systems could also be exploited for enhanced sensing.

**Nonreciprocal sensors and the mode-splitting technique.** Up to this point, our work has focused exclusively on sensing parametric changes in $\epsilon$ that are small enough to allow the use of a perturbative, linear response approach; this typically requires $\epsilon$ to be smaller than relevant mode linewidths. Nonreciprocity enhanced sensing is however also highly effective for larger, nonperturbative changes in $\epsilon$. We consider the same general setting as recent works on EP sensing[10,13,19,20] that aim to detect a relatively large change in $\epsilon$ by directly measuring the frequency splitting of two normal modes[42–44]. This involves first measuring the output field intensity as a function of drive frequency, and then fitting this curve to extract a mode splitting.

For a sufficiently strong classical drive the contribution of amplified vacuum fluctuations can be ignored, and the intensity of the waveguide output field $\hat{B}^{\text{out}}$ (c.f., Eq. (9)) is

$$\begin{aligned} \mathcal{P}[\Delta] &\equiv \left\langle \left(\hat{B}^{\text{out}}\right)^\dagger \hat{B}^{\text{out}} \right\rangle \\ &\approx \left\langle \hat{B}^{\text{out}} \right\rangle^* \left\langle \hat{B}^{\text{out}} \right\rangle = \beta^2 \left|1 - \chi_{11}\right|^2. \end{aligned} \quad (39)$$

where $\Delta$ is as always the detuning of the drive frequency from the cavity 1 resonance frequency, and $\beta$ the (real) amplitude of incident driving field.

As discussed before, the magnitude of $\chi$ is large when $\Delta$ is close to an eigenfrequency of $\tilde{H}$, hence $\mathcal{P}[\Delta]$ will generically exhibit a resonance feature (peak or dip) near these values. If a nonzero $\epsilon$ lifts the degeneracy of eigenvalues, it will thus manifest itself by the appearance of new resonances in the intensity spectrum. We note again that the transmitted field in standard setups where a nearby readout object (e.g., prism or fiber) is coupled to an optical resonator[20,42,44,45] is completely equivalent to the reflected field in our geometry.

As we now show, nonreciprocity has two distinct benefits to frequency-splitting detection. First, a perturbation to a nonreciprocal system can induce new resonances in $\mathcal{P}[\Delta]$ even if there is no degeneracy in the unperturbed system. This allows the frequency-splitting technique to be implemented in a wider range of systems. Second, nonreciprocity can dramatically increase the parametric, $\epsilon$-dependent splitting of resonances: one can obtain the same $\sqrt{\epsilon}$ type splitting as a system tuned to an EP, without actually needing to be at an EP. Again, this greatly reduces the fine tuning needed to achieve such strong parametric mode splittings (and also demonstrates that EP is not a necessary ingredient for such behavior).

Consider the first point above: with nonreciprocity, the mode-splitting technique can be used even if the unperturbed system has non-degenerate eigenvalues. The reason is simple: because of nonreciprocity, a given eigenmode of the system may fail to be excited by the incident measurement drive when $\epsilon = 0$, irrespective of $\Delta$. If however a nonzero $\epsilon$ breaks the system's nonreciprocity, these dark modes may become visible in the output intensity spectrum. Further, breaking nonreciprocity can lead to parametrically large mode splittings, much larger than would be possible without nonreciprocity.

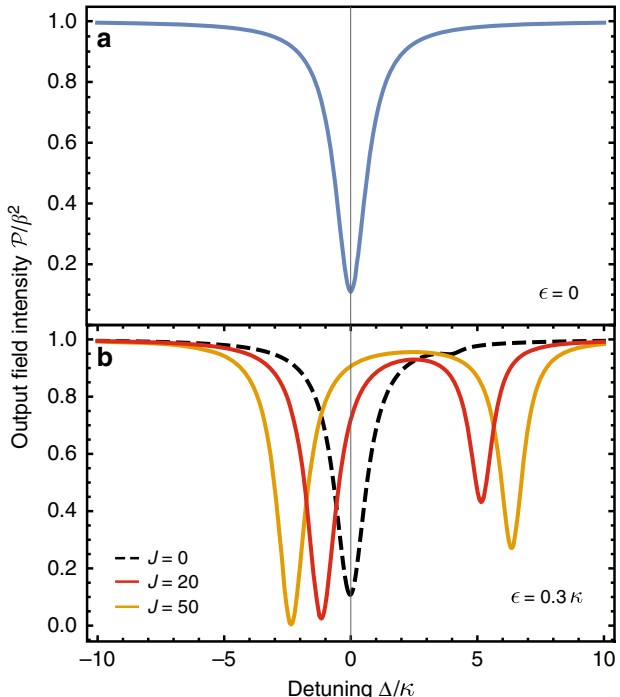

**Fig. 6** Drive-detuning dependence of the output field intensity. The dependence on of the output field intensity $\mathcal{P}[\Delta]$ (c.f., Eq. (39)) for a nonreciprocal system is described by Eq. (35). We have taken $\gamma_1 = 0.5\kappa$, $\gamma_2 = \kappa$, $\nu_2 = 4\kappa$. **a** Spectrum for $\epsilon = 0$, when the system is fully nonreciprocal. Even though the system has two nondegenerate eigenvalues $\Omega_\pm[0]$, only one resonance is seen, as nonreciprocity makes the $\Omega_+[0]$ dark to the incident drive. Note that this spectrum is independent of $J$. **b** Black dashed: Spectrum where the parameter to be sensed $\epsilon = 0.3\kappa$, but without nonreciprocity ($J = 0$). The spectrum only has a single dip, and almost identical to the $\epsilon = 0$ spectrum. Red and green: spectra with the same perturbation $\epsilon = 0.3\kappa$, but with nonreciprocal couplings $J = 20\kappa$ and $J = 50\kappa$, respectively. Two resonances are clearly observed, and their separation increases with the strength of the nonreciprocal coupling $J$

To illustrate both the above points, we again consider the simple two-mode nonreciprocal sensor described by Eq. (35). As usual, the parameter to be sensed corresponds to a Hermitian coupling between the two modes (c.f., Eq. (26)). For $\epsilon = 0$, we have a purely nonreciprocal coupling between the modes: mode 2 can influence mode 1, but not vice-versa. One finds that the eigenvalues of $\tilde{H}_{\mathrm{dir}}[0]$, as given in Eq. (37), are in general nondegenerate in both real and imaginary parts. Note also that the eigenvalues are completely independent of the coupling $J$, reflecting the lack of any coherent oscillations between mode 1 and 2.

When perturbation $\epsilon$ is nonzero, reciprocity is lost, as mode 1 can now influence mode 2. The eigenvalues of $\tilde{H}_{\mathrm{dir}}[\epsilon]$ are

$$\Omega_\pm[\epsilon] = \frac{\nu_2}{2} - i\frac{\kappa + \gamma_1 + \gamma_2}{4}$$
$$\pm \sqrt{J\frac{\epsilon}{2} + \frac{\epsilon^2}{4} + \left(\frac{\nu_2}{2} + i\frac{\kappa + \gamma_1 - \gamma_2}{4}\right)^2}. \quad (40)$$

For $\epsilon > 0$ and sufficiently large positive $J$, the frequency splitting demonstrates a square root dependence on $\epsilon$, i.e., $\Omega_+[\epsilon] - \Omega_-[\epsilon] \approx \sqrt{2J\epsilon}$. We see that the nonreciprocal coupling $J$ amplifies the effect of $\epsilon$, even though it has no impact on the unperturbed eigenvalues. The $\sqrt{\epsilon}$ splitting dependence resembles that of EP sensing schemes; here, however, the unperturbed modes are not required to be tuned to a degeneracy, and the unperturbed system is not at an EP. The enhanced splitting

obtained here can be much larger than mode linewidths, and directly manifests itself in the output intensity spectrum $\mathcal{P}[\Delta]$. An example is shown in Fig. 6.

## Discussion

We have provided a comprehensive analysis of weak dispersive-style measurements made using coupled-mode systems described by effective non-Hermitian Hamiltonians. Our work goes beyond previous analyses of non-Hermitian sensing techniques, in that we fully treat fluctuation effects, and fully treat the entire measurement process. We derive fundamental bounds on any reciprocal two-mode non-Hermitian sensor, and show that they also constrain systems that are tuned to an exceptional point. Generically, we find that amplification is the necessary ingredient for generating large signal powers, and this can be achieved without any proximity to an exceptional point. However, amplification process must incorporate extra noise that will fundamentally limit the quantum measurement rate of any reciprocal sensor. Our results highlight the fact that the efficacy of a non-Hermitian sensing scheme is not completely described by the parametric dependence of mode eigenvalues. Considering fluctuation effects, the particular dissipative implementation of the dynamics is crucial as this will set noise levels.

We also discussed a new method for enhancing dispersive measurement using effective non-Hermitian physics, namely the use of nonreciprocity to enhance sensing. We show that nonreciprocity allows one to arbitrarily exceed the fundamental bound on the measurement rate of a reciprocal sensor, and discussed a simple implementation that does not require any amplification processes. We also show that nonreciprocity can enhance the sensitivity of mode-splitting type sensor.

Finally, we note that the general theory developed in this work could be easily applied to more general kinds of sensing problems. For example, the same formalism could be used to understand the performance of non-Hermitian sensors when thermal noise dominates (as would be the case for systems deep in the classical limit). The formalism could also be extended to study the sensing of time-dependent perturbations.

Note added: During the completion of this work, we became aware of work by Zhang et al.[49] and Chen et al.[50] on a related topic.

## Methods

**Full Hamiltonian for effective non-Hermitian system.** Our measurement setup consists of a readout waveguide that interacts with only one cavity mode. This coupled cavity mode can interact with other modes as well as arbitrary gain and loss baths. The interaction between cavities is limited to be photon number conserving, i.e., only hopping. The total Hamiltonian of the system + bath is

$$
\begin{aligned}
\hat{\mathbb{H}} = &\sum_{i,j=1}^{M} H_{ij}\hat{a}_i^\dagger \hat{a}_j + \int dk \left(\omega_{b,k}\hat{b}_k^\dagger \hat{b}_k\right) \\
&+ \sum_{j=1}^{N_Y}\int dk\left(\omega_{c,j,k}\hat{c}_{j,k}^\dagger \hat{c}_{j,k}\right) \\
&+ \sum_{j=1}^{N_Z}\int dk\left(\omega_{d,j,k}\hat{d}_{j,k}^\dagger \hat{d}_{j,k}\right) \\
&+ \int \frac{dk}{\sqrt{\pi}}g(k)\left(\hat{a}_1\hat{b}_k^\dagger + \hat{a}_1^\dagger \hat{b}_k\right) \\
&+ \sum_{i=1}^{M}\sum_{l=j}^{N_Y}\int \frac{dk}{\sqrt{\pi}}\left(Y_{ij}^*(k)\hat{a}_i\hat{c}_{j,k} + Y_{ij}(k)\hat{a}_i^\dagger \hat{c}_{j,k}^\dagger\right) \\
&+ \sum_{i=1}^{M}\sum_{j=1}^{N_Z}\int \frac{dk}{\sqrt{\pi}}\left(Z_{ij}^*(k)\hat{a}_i\hat{d}_{j,k}^\dagger + Z_{ij}(k)\hat{a}_i^\dagger \hat{d}_{j,k}\right).
\end{aligned}
\quad (41)
$$

$\hat{b}_k$, $\hat{c}_{j,k}$, and $\hat{d}_{j,k}$ are the annihilation operator of the mode with wave number $k$ in the readout waveguide, gain bath, and loss bath, respectively. Mode operators of different bath commute, and that of the same bath follows $\left[\hat{O}_k, \hat{O}_{k'}^\dagger\right] = \delta(k - k')$, where $O \in \{b_k, c_{j,k}, d_{j,k}\}$. We have chosen a unit that $\hbar = 1$, and $k$ has a unit of frequency. For simplicity, we assume all baths have linear dispersion relation, i.e.,

$\omega_{b,k} = \omega_{c,j,k} = \omega_{d,j,k} = k$, and homogeneous mode-bath coupling, i.e., $g(k) = \sqrt{\kappa}$, $Y_{ij}(k) = Y_{ij}$ and $Z_{ij}(k) = Z_{ij}$.

With the full Hamiltonian in Eq. (41), the Heisenberg equations of motion (HEOM) of any operator $\hat{A}$ can be computed as $\dot{\hat{A}} = i[\hat{\mathbb{H}}, \hat{A}]$. By integrating the HEOM of bath mode operators, and substituting them into the HEOM of cavity mode operators, the Langevin equation in Eq. (5) can be arrived at. In our convention, the input field operators are defined as

$$\hat{B}^{\text{in}}(t) = \int \frac{dk}{\sqrt{2\pi}} \hat{b}_k(t_0) e^{-ik(t-t_0)} \tag{42}$$

$$\hat{C}_j^{\text{in}}(t) = \int \frac{dk}{\sqrt{2\pi}} \hat{c}_{j,k}(t_0) e^{-ik(t-t_0)} \tag{43}$$

$$\hat{D}_j^{\text{in}}(t) = \int \frac{dk}{\sqrt{2\pi}} \hat{d}_{j,k}(t_0) e^{-ik(t-t_0)}, \tag{44}$$

where $t_0 \to -\infty$. Interested readers can refer to refs.[5,23] for detailed derivations.

**Minimum noise.** For a given $\tilde{H}$, the measurement noise depends on the choice of gain and loss baths. We can optimize the choice to obtain a minimum measurement noise. We first recognize in Eq. (23) that $(\chi YY^\dagger \chi^\dagger)_{11} \geq 0$, because $YY^\dagger$ is positive semidefinite. By using Eq. (4), we can obtain another relation

$$(\chi YY^\dagger \chi^\dagger)_{11} = \frac{1}{2i}\left((\chi\tilde{H}\chi^\dagger)_{11} - (\chi\tilde{H}^\dagger\chi^\dagger)_{11}\right) + \frac{\kappa}{2}|\chi_{11}|^2 + (\chi ZZ^\dagger\chi^\dagger)_{11} \tag{45}$$

$$\geq -\frac{\kappa}{2}(\chi_{11}^* + \chi_{11}) + \frac{\kappa}{2}|\chi_{11}|^2, \tag{46}$$

where we employ the fact that $ZZ^\dagger$ is positive semidefinite in the last relation. After rearrangement, we get the minimum noise in Eq. (46).

**Single-tone quantum Fisher information.** We want to characterize the maximum amount of information available on $\epsilon$ in the reflected output mode in our waveguide. As we are interested in the limit of long integration times $\tau$, the relevant temporal mode of the output field is described by an annihilation operator

$$\hat{\mathcal{B}}(\tau) \equiv \frac{1}{\sqrt{\tau}} \int_0^\tau \hat{B}^{\text{out}}(t) dt. \tag{47}$$

This is a standard bosonic annihilation operator satisfying $[\hat{\mathcal{B}}(\tau), \hat{\mathcal{B}}^\dagger(\tau)] = 1$.

Changing the parameter $\epsilon$ will change both our non-Hermitian Hamiltonian $\tilde{H}$ as well as the state of the temporal mode $\hat{\mathcal{B}}$. To sense this change, one would measure some property of $\hat{\mathcal{B}}$, described by an observable $\hat{\mathcal{M}}$. The possible outcomes $z$ of the measurement would be described by a probability distribution $P_\epsilon[z]$, which depends parametrically on $\epsilon$. Our goal is to maximize the the statistical distance between $P_\epsilon[z]$ and $P_0[z]$. For small $\epsilon$, standard definitions and arguments yield that this distance $ds^2$ is given by $\epsilon^2\mathcal{F}$, where $\mathcal{F}$ is the Fisher information[46]. Optimizing $\mathcal{F}$ over all possible choices of measurement observables $\hat{\mathcal{M}}$ yields the quantum Fisher information (QFI), $\mathcal{F}_{\text{QFI}}$[24,47].

In our case, because of the linear nature of our system and the Gaussian nature of the relevant noise, $\hat{\mathcal{B}}$ is always in a Gaussian state, and $\mathcal{F}_{\text{QFI}}$ can be computed exactly[25]. For infinitesimal $\epsilon$, one finds:

$$\mathcal{F}_{\text{QFI}} = \left(\frac{d\vec{u}_\epsilon}{d\epsilon} W_\epsilon^{-1} \frac{d\vec{u}_\epsilon^{\text{T}}}{d\epsilon}\right)\bigg|_{\epsilon\to 0} + \Xi, \tag{48}$$

where $\vec{u}_\epsilon \equiv (\langle\hat{q}_1\rangle_\epsilon, \langle\hat{q}_2\rangle_\epsilon)$ and $(W_\epsilon)_{jl} \equiv \frac{1}{2}\langle\{\delta\hat{q}_j, \delta\hat{q}_l\}\rangle_\epsilon$ are, respectively the first and second moments of the Gaussian state; $\hat{q}_1 \equiv (\hat{\mathcal{B}} + \hat{\mathcal{B}}^\dagger)/\sqrt{2}$ and $\hat{q}_2 \equiv i(-\hat{\mathcal{B}} + \hat{\mathcal{B}}^\dagger)/\sqrt{2}$ are the $\mathcal{B}$ mode quadratures; $\delta\hat{q}_j \equiv \hat{q}_j - \langle\hat{q}_j\rangle_\epsilon$[48]. $\Xi$ is a scalar that depends on only the $\epsilon$-dependence of the second moment. The first (second) term in Eq. (48) can be viewed as the information associated with the $\epsilon$-induced change in the first (second) moment of the temporal mode $\mathcal{B}$.

After solving the Langevin equation in Eq. (5), the first and second moment for our linear sensor can be evaluated in the long-$\tau$ limit of interest:

$$\vec{u}_\epsilon = \sqrt{2\tau}\beta\left(1 - \text{Re}\,\tilde{\chi}_{11}[0;\Delta;\epsilon], -\text{Im}\,\tilde{\chi}_{11}[0;\Delta;\epsilon]\right) \tag{49}$$

$$W_0 = \frac{\bar{S}_{II}[0]}{\kappa}\begin{pmatrix} 1 & 0 \\ 0 & 1 \end{pmatrix} \tag{50}$$

Due to the linearity of Eqs. (5) and (9), the classical drive affects only the first but not the second moment of the output field. This can be seen from the fact that the first moment in Eq. (49) scales as $\beta$, while the second moment in Eq. (50) is independent

of $\beta$. For sufficiently strong drive, the QFI will be dominated by the first, drive-dependent term in Eq. (48), and the contribution from $\Xi$ can be neglected.

One can now confirm that the SNR for an optimal homodyne measurement (as given in in Eq. (16)) coincides with the quantum Fisher information, i.e., $\text{SNR} = \epsilon^2\mathcal{F}_{\text{QFI}}$. This implies that homodyne detection is the optimal measurement for dispersive sensing because it extracts the maximum information about $\epsilon$ from $\mathcal{B}$ mode.

**Multiple-tone quantum Fisher information.** In the main text and previous sections, we considered the case where the system is driven at a single frequency. One might naturally ask if the measurement rate can be increased by driving and measuring the system using multiple drive tones, each at a different frequency. For sufficiently many frequencies, this method is equivalent to probing the full spectral response of the system.

As mentioned in the main text, this multitone approach is no better that simply probing the system with a single tone with an optimally chosen driving frequency. We make this statement rigorous here. We here show that if the total intracavity photon number is restricted, then probing the entire spectral response (via multitone driving) does not provide more information (as quantified by the quantum Fisher information) than an optimal single-tone measurement.

We first consider a generalized coherent driving field on mode 1 that consists of $N_B$ distinct frequencies:

$$\beta(t) = \sum_{j=1}^{N_B} \beta_j e^{-i\Delta_j t}. \tag{51}$$

In the long-time limit, the output field state becomes a dynamic steady state that consists of components in each tone $\Delta_j$. Each component can be viewed as the state of a temporal mode:

$$\hat{\mathcal{B}}_j \equiv \frac{1}{\sqrt{\tau}} \int_0^\tau \hat{B}^{\text{out}}(t) e^{i\Delta_j t} dt. \tag{52}$$

It is easy to check that temporal modes are independent bosonic modes at $\tau \to \infty$, i.e., $[\hat{\mathcal{B}}_j, \hat{\mathcal{B}}_l] = 0$ and $[\hat{\mathcal{B}}_j, \hat{\mathcal{B}}_l^\dagger] = \delta_{jl}$.

Because the system is linear, the multimode output state is Gaussian. We again assume each drive is sufficiently strong that QFI is dominated by the first term in Eq. (48). The multimode first moment is $\vec{u}_\epsilon = \left(\langle\hat{q}_{1,1}\rangle_\epsilon, \langle\hat{q}_{2,1}\rangle_\epsilon, \langle\hat{q}_{1,2}\rangle_\epsilon, \langle\hat{q}_{2,2}\rangle_\epsilon, \cdots\right)$, where the quadrature operators of each mode are $\hat{q}_{1,j} \equiv (\hat{\mathcal{B}}_j + \hat{\mathcal{B}}_j^\dagger)/\sqrt{2}$ and $\hat{q}_{2,j} \equiv i(-\hat{\mathcal{B}}_j + \hat{\mathcal{B}}_j^\dagger)/\sqrt{2}$. The first moment of each mode can be evaluated by

$$\langle\hat{\mathcal{B}}_j\rangle_\epsilon = \sqrt{\tau}\beta_j\left(1 - \tilde{\chi}_{11}[0;\Delta_j;\epsilon]\right). \tag{53}$$

In the long-time limit, we find that the second moment is block diagonal, i.e., $W_0 = \oplus_{j=1}^{N_B} W_0^{(j)}$, and each block corresponds to the second moment of each temporal mode:

$$W_0^{(j)} = \frac{\bar{S}_{II}[\Delta_j]}{\kappa}\begin{pmatrix} 1 & 0 \\ 0 & 1 \end{pmatrix} \tag{54}$$

where

$$\bar{S}_{II}[\Delta_j] = \frac{\kappa}{2}\left(1 + \frac{4}{\kappa}\left(\chi(\Delta_j)YY^\dagger\chi^\dagger(\Delta_j)\right)_{11}\right), \tag{55}$$

and $\chi(\Delta_j) \equiv \tilde{\chi}[0;\Delta_j;0]$. Note that $\bar{S}_{II}[\Delta] \equiv S_{II}[0]$ in Eq. (23) because all dynamics in the main text is evaluated at the rotating frame of the single drive frequency.

To fairly compare this multitone approach with other schemes, we again constrain the problem to have a fixed total photon number. Here the time-averaged total photon number is

$$\bar{n}_{\text{tot}} = \sum_{j=1}^{N_B} \bar{n}_j \equiv \sum_{j=1}^{N_B} \frac{\beta_j^2}{\kappa}\left(\chi^\dagger(\Delta_j)\chi(\Delta_j)\right)_{11}. \tag{56}$$

The multitone Fisher information can be evaluated as

$$\mathcal{F}_{\text{mt}} = \sum_{j=1}^{N_B} \frac{\tau}{\kappa^2}\bar{n}_j\tilde{\Gamma}(\Delta_j) \tag{57}$$

where $\tilde{\Gamma}(\Delta_j)$ is the measurement rate per coherent photon for a single-tone measurement at detuning $\Delta_j$:

$$\tilde{\Gamma}(\Delta_j) \equiv \frac{2\kappa^2}{\bar{S}_{II}[\Delta_j]} \frac{\left|\left(\chi(\Delta_j)V\chi(\Delta_j)\right)_{11}\right|^2}{\left(\chi^\dagger(\Delta_j)\chi(\Delta_j)\right)_{11}}. \tag{58}$$

 

By using Eq. (46), it is easy to show that each $S_{II}[\Delta_j]$ is lower bounded by

$$
\begin{aligned}
\tilde{S}_{II}\left[\Delta_j\right] & \geq \tilde{S}_{II}\left[\Delta_j\right]_{\min} \\
& \equiv \frac{\kappa}{2}\left(1 + 2\Theta\left(\left|1 - \chi_{11}\left(\Delta_j\right)\right|^2 - 1\right)\right. \\
& \left. \times\left(\left|1 - \chi_{11}\left(\Delta_j\right)\right|^2 - 1\right)\right)
\end{aligned}
\tag{59}
$$

and so the maximum per-photon measurement rate is

$$
\tilde{\Gamma}\left(\Delta_j\right) \leq \tilde{\Gamma}_{\mathrm{opt}}\left(\Delta_j\right) \equiv \frac{2\kappa^2}{\tilde{S}_{II}\left[\Delta_j\right]_{\min}} \frac{\left|\left(\chi\left(\Delta_j\right) V \chi\left(\Delta_j\right)\right)_{11}\right|^2}{\left(\chi^\dagger\left(\Delta_j\right) \chi\left(\Delta_j\right)\right)_{11}}.
\tag{60}
$$

We note that there might not be a single set of bath that could optimize all $\tilde{S}_{II}[\Delta_j]$ to saturate the bound in Eq. (59). In general the bath can only be optimized with respect to a specific detuning $\Delta_j$.

We can then see that the multitone QFI is upper-bounded by the maximum single-tone QFI at the optimal detuning:

$$
\begin{aligned}
\mathcal{F}_{\mathrm{mt}} & \leq \sum_{j=1}^{N_B} \frac{\tau}{\kappa^2} \bar{n}_j \tilde{\Gamma}_{\mathrm{opt}}\left(\Delta_j\right) \\
& \leq \sum_{j=1}^{N_B} \frac{\tau}{\kappa^2} \bar{n}_j \max_{\Delta_j}\left(\tilde{\Gamma}_{\mathrm{opt}}\right) = \frac{\tau}{\kappa^2} \bar{n}_{\mathrm{tot}} \max_{\Delta_j}\left(\tilde{\Gamma}_{\mathrm{opt}}\right) \\
& = \max_{\Delta_j}\left(\mathcal{F}_{\mathrm{QFI}}\right).
\end{aligned}
\tag{61}
$$

Recall again that the QFI is the maximum information obtainable from any detection scheme. Our result thus shows that probing the entire spectral response of the system (via multitone driving) does not provide more information about the parameter $\epsilon$ than simply driving with a single (optimally chosen) tone and performing a homodyne measurement.

**Bound on the measurement rate of a reciprocal two-mode system.** We focus here on a two-mode system where the parameter to be sensed corresponds to the Hermitian coupling between the modes (as given in Eq. (26) in the main text). For any such two-mode system, the maximum measurement rate obtainable by using an optimized bath is

$$
\Gamma_{\mathrm{meas}} \leq \Gamma_{\mathrm{opt}} = \kappa \bar{n}_{\mathrm{tot}} \frac{\left|\chi_{12} + \chi_{21}\right|^2}{\left|\chi_{11}\right|^2 + \left|\chi_{21}\right|^2} f\left(\chi_{11}\right)
\tag{62}
$$

where $f(\chi_{11})$ is a positive-valued function of a complex $\chi_{11}$:

$$
f\left(\chi_{11}\right) \equiv \frac{\left|\chi_{11}\right|^2}{1 + 2\Theta\left[\left|1 - \chi_{11}\right|^2 - 1\right]\left(\left|1 - \chi_{11}\right|^2 - 1\right)}.
\tag{63}
$$

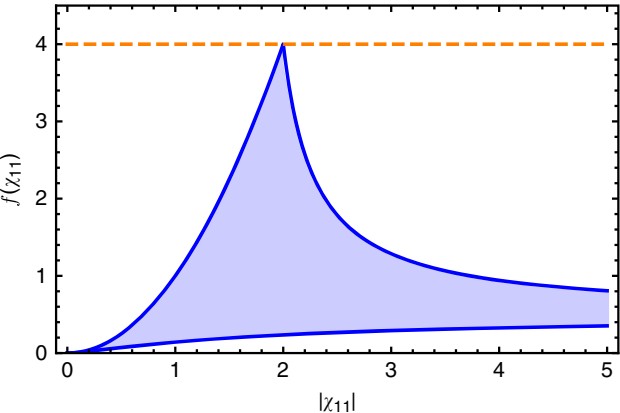

**Fig. 7** Allowable values of function $f(\chi_{11})$. The shaded blue region indicates the allowable values of $f(\chi_{11})$ (c.f., Eq. (63)) as a function $|\chi_{11}|$. $f(\chi_{11})$ sets the maximum possible measurement rate both for sensing a parametric change in mode coupling (c.f., Eq. (62)) and for sensing a parametric change in the resonance frequency of mode 1 (c.f., Eq. (67)). The dashed horizontal line shows the maximum possible value of $f(\chi_{11})$, which is achieved only when $\chi_{11} = 2$. Note that for larger $|\chi_{11}| \geq 5$, $f(\chi_{11}) < 1$

Due to reciprocity, i.e., $|\chi_{12}| = |\chi_{21}|$, the sum of antidiagonal susceptibility entries is bounded by $\left|\chi_{12} + \chi_{21}\right|^2 \leq 4\left|\chi_{21}\right|^2$. This condition bounds the optimal measurement rate as

$$
\Gamma_{\mathrm{opt}} \leq 4\kappa \bar{n}_{\mathrm{tot}} f\left(\chi_{11}\right) \leq 16\kappa \bar{n}_{\mathrm{tot}}.
\tag{64}
$$

The first inequality is exploited when $\left|\chi_{21}\right|^2 \gg \left|\chi_{11}\right|^2$. The second inequality is imposed by the maximum value of $f(\chi_{11})$. As illustrated in Fig. 7, the maximum is $\max\{f(\chi_{11})\} = 4$, which is attainable when $\chi_{11} = 2$. These inequalities complete the reciprocal-system bound in Eq. (30).

**Bounds on sensing a change in the frequency of mode 1.** Consider a general $M$ mode setup where the parameter of interest is a simple shift in the resonance frequency of mode 1, i.e., $V_{ij} = \delta_{i1}\delta_{j1}$. In this case, one finds from Eq. (21) that the signal power $\mathcal{S}$ is given by

$$
\mathcal{S} = \frac{1}{4} \frac{\left|\chi_{11}\right|^4}{\left(\chi^\dagger \chi\right)_{11}} \mathcal{S}_e \leq \frac{1}{4}\left|\chi_{11}\right|^2 \mathcal{S}_e,
\tag{65}
$$

where $\mathcal{S}_e$ is defined in Eq. (28). The last relation becomes an equality in the special case where $\chi_{j1} = 0$ for $j \neq 1$. In this case, the coherent drive only induces a coherent photon population in mode 1 (and not in modes 2 through $M$).

If we further specialize to a system with just one mode ($M = 1$), the Hamiltonian and susceptibility are simple scalars,

$$
\tilde{H}_{\mathrm{one}}[0] = -i\frac{\kappa + \gamma_1}{2}, \quad \chi_{11} = \frac{i\kappa}{\Delta + i(\kappa + \gamma_1)/2}.
\tag{66}
$$

If we further assume a resonant drive ($\Delta = 0$) and no extra gain or loss ($\gamma_1 = 0$), we have the usual setup for an ideal dispersive measurement (see, e.g., Ref.[5]). One has $\chi_{11} = 2$, implying that the signal power is just $\mathcal{S} = \mathcal{S}_e$.

If one now allows for gain (i.e., $\gamma_1 < 0$), the signal power can be enhanced arbitrarily above $\mathcal{S}_e$ without increasing the intracavity photon number $\bar{n}_{\mathrm{tot}}$: one simply increases $\left|\chi_{11}\right|$ above 2. We stress that this enhancement does not require EP, but simply the introduction of gain, and also applies to the case of a multimode system $M > 1$.

Consider next the behavior of the measurement rate $\Gamma_{\mathrm{meas}}$ associated with detecting a mode-1 frequency shift; as discussed, $\Gamma_{\mathrm{meas}}$ considers both the signal power and the impact of intrinsic noise in the homodyne current. For a general $M$ mode sensor, we find that $\Gamma_{\mathrm{meas}}$ is upper-bounded by

$$
\Gamma_{\mathrm{meas}} \leq 4\kappa \bar{n}_{\mathrm{tot}} f\left(\chi_{11}\right) \leq 16\kappa \bar{n}_{\mathrm{tot}},
\tag{67}
$$

where $f(\chi_{11})$ is defined in Eq. (63). As shown in Fig. 7, the measurement rate is maximum when $\chi_{11} = 2$. This optimal value is achieved by the simple one mode sensor in Eq. (66) in the case where the dissipation of mode 1 is solely due to the waveguide coupling, i.e., $\gamma_1 = 0$. It is interesting to note that this bound is identical to the bound on a reciprocal two-mode sensor, c.f., Eq. (30). In contrast, a nonreciprocal two-mode sensor could have $\Gamma_{\mathrm{meas}}$ arbitrarily larger than this bound, c.f., Eq. (34).

Because a simple one-mode system achieves the optimal value of $\Gamma_{\mathrm{meas}}$ for sensing a parametric change in the frequency of mode 1, using a multimode system is unnecessary for this problem. This is true even if one uses a multimode system tuned to an EP where eigenvalues exhibit a $\sqrt{\epsilon}$ scaling. As a concrete example, consider the reciprocal two-mode system given in Eq. (31). When $J = (\kappa + \gamma_1 - \gamma_2)/4$, the system exhibits EP. The unscaled Jordan normal form can be obtained in an appropriate basis:

$$
\frac{1}{2}\begin{pmatrix} 1 & -i \\ -i & 1 \end{pmatrix} \tilde{H}_{\mathrm{recip}}[0] \begin{pmatrix} 1 & i \\ i & 1 \end{pmatrix} = \begin{pmatrix} \Omega[0] & 2J \\ 0 & \Omega[0] \end{pmatrix},
\tag{68}
$$

where the unperturbed degenerate eigenvalue is $\Omega[0] \equiv -i(\kappa + \gamma_1 + \gamma_2)/4$. In this basis, the perturbation matrix has non-vanishing off-diagonal entry, i.e.,

$$
\frac{1}{2}\begin{pmatrix} 1 & -i \\ -i & 1 \end{pmatrix} V \begin{pmatrix} 1 & i \\ i & 1 \end{pmatrix} = \begin{pmatrix} 1/2 & i/2 \\ -i/2 & 1/2 \end{pmatrix},
\tag{69}
$$

and so the eigenvalue of $\tilde{H}[\epsilon]$ has $\sqrt{\epsilon}$ dependence at small $\epsilon$, i.e.,

$$
\Omega[\epsilon] \approx \Omega[0] \pm \sqrt{-i\epsilon J}.
\tag{70}
$$

As usual, one might be tempted to conclude from this strong dependence of eigenvalues on $\epsilon$ that this system should out perform the simple one-mode system in Eq. (66). This is, however, not true: the two-mode EP system still has a measurement rate fundamentally bounded by Eq. (67). It is interesting to note that this bound is achieved by the two-mode EP system only when mode 2 has nonzero loss, i.e., $\gamma_2 > 0$, and when the gain of mode 1 is tuned to $\gamma_1 = -\left(\sqrt{\kappa} - \sqrt{\gamma_2}\right)^2$.

**Systematic construction of minimum noise bath.** In Eq. (46), we show the lower bound of measurement noise for a given $\tilde{H}$. Here, we outline a systematic

 

construction of baths that attains the minimum measurement noise. We first recall that the coupling to gain and loss bath is specified by the positive semidefinite $M \times M$ matrices $YY^\dagger$ and $ZZ^\dagger$. For a given $\tilde{H}$, those matrices are not unique because Eq. (4) is unchanged if we change the baths as $YY^\dagger \to YY^\dagger + K$ and $ZZ^\dagger \to ZZ^\dagger + K$, for any positive semidefinite $K$. Our aim is to find a $YY^\dagger$ such that the gain noise term is

$$\left(\chi YY^\dagger \chi^\dagger\right)_{11} = \max\left\{\left(\chi\left(\frac{\tilde{H} - \tilde{H}^\dagger + i\tilde{\kappa}}{2i}\right)\chi^\dagger\right)_{11}, 0\right\}. \quad (71)$$

For convenience, we define the Hermitian matrix

$$h \equiv \chi\left(\frac{\tilde{H} - \tilde{H}^\dagger + i\tilde{\kappa}}{2i}\right)\chi^\dagger. \quad (72)$$

Then Eq. (71) becomes a conditional equation of $h_{11}$ only. In the following, we separately consider the cases of $h_{11} < 0$ and $h_{11} > 0$.

For negative $h_{11}$, our aim is to construct $YY^\dagger$ such that

$$\left(\chi YY^\dagger \chi^\dagger\right)_{11} = 0. \quad (73)$$

We first construct a positive semidefinite Hermitian matrix $X_{-1}$:

$$(X_{-1})_{1i} = (X_{-1})^*_{i1} = -h_{1i} = -h^*_{i1}, \quad (74)$$

$$(X_{-1})_{jj} = -\frac{M|h_{1j}|^2}{h_{11}}, \quad (75)$$

$$(X_{-1})_{ij} = 0 \quad \text{otherwise}. \quad (76)$$

It is easy to see that $h + X_{-1}$ has vanishing entries in first row and first column, i.e., $(h + X_{-1})_{1i} = (h + X_{-1})_{i1} = 0$.

Because $h + X_{-1}$ is Hermitian, it can always be diagonalized by a unitary matrix $U$:

$$\left(U(h + X_{-1})U^\dagger\right)_{ij} = \Lambda_i \delta_{ij}, \quad (77)$$

where the eigenvalues $\Lambda_i$ are real. Due to the vanishing first row and first column in $h + X_{-1}$, we can require $U_{1i} = U_{i1} = \delta_{i1}$ and $\Lambda_1 = 0$.

Next we decompose $h + X_{-1} = X_{+2} - X_{-2}$ as the difference of two positive semidefinite Hermitian matrices $X_{+2}$ and $X_{-2}$, which are constructed as

$$\left(UX_{\pm2}U^\dagger\right)_{ij} \equiv \frac{|\Lambda_i| \pm \Lambda_i}{2}\delta_{ij}. \quad (78)$$

Combining the matrices and Eq. (72), we have

$$\chi\left(\frac{\tilde{H} - \tilde{H}^\dagger + i\tilde{\kappa}}{2i}\right)\chi^\dagger = X_{+2} + (-X_{-1} - X_{-2}), \quad (79)$$

where the first (second) term in R.H.S. is positive (negative) semidefinite and thus corresponds to gain (loss) bath. By using Eqs. (4) and (17), we can construct the gain and loss baths as

$$YY^\dagger = \frac{1}{\kappa^2}(\Delta\mathbb{I} - \tilde{H}[0])X_{+2}(\Delta\mathbb{I} - \tilde{H}^\dagger[0]), \quad (80)$$

$$ZZ^\dagger = \frac{1}{\kappa^2}(\Delta\mathbb{I} - \tilde{H}[0])(X_{-1} + X_{-2})(\Delta\mathbb{I} - \tilde{H}^\dagger[0]). \quad (81)$$

It is easy to verify that Eq. (73) is satisfied.

Similarly for positive $h_{11}$, our aim is to construct $YY^\dagger$ such that

$$\left(\chi YY^\dagger \chi^\dagger\right)_{11} = h_{11}. \quad (82)$$

We first construct a positive semidefinite Hermitian matrix $X_{+1}$:

$$(X_{+1})_{1i} = (X_{+1})^*_{i1} = h_{1i} = h^*_{i1}, \quad (83)$$

$$(X_{+1})_{jj} = \frac{M|h_{1j}|^2}{h_{11}}, \quad (84)$$

$$(X_{+1})_{ij} = 0 \quad \text{otherwise}. \quad (85)$$

It is easy to see that $h - X_{+1}$ has vanishing entries in first row and first column, i.e., $(h - X_{+1})_{1i} = (h - X_{+1})_{i1} = 0$.

Because $h - X_{+1}$ is Hermitian, it can always be diagonalized by a unitary matrix $U$:

$$\left(U(h - X_{+1})U^\dagger\right)_{ij} = \Lambda_i \delta_{ij}, \quad (86)$$

where the eigenvalues $\Lambda_i$ are real. Due to the vanishing first row and first column in $h - X_{+1}$, we can require $U_{1i} = U_{i1} = \delta_{i1}$ and $\Lambda_1 = 0$.

Next we decompose $h - X_{+1} = X_{+2} - X_{-2}$ as the difference of two positive semidefinite Hermitian matrices $X_{+2}$ and $X_{-2}$, which are constructed as

$$\left(UX_{\pm2}U^\dagger\right)_{ij} \equiv \frac{|\Lambda_i| \pm \Lambda_i}{2}\delta_{ij}. \quad (87)$$

Combining the matrices and Eq. (72), we have

$$\chi\left(\frac{\tilde{H} - \tilde{H}^\dagger + i\tilde{\kappa}}{2i}\right)\chi^\dagger = (X_{+1} + X_{+2}) + (-X_{-2}), \quad (88)$$

where the first (second) term in R.H.S. is positive (negative) semidefinite and thus corresponds to gain (loss) bath. By using Eqs. (4) and (17), we can construct the gain and loss baths as

$$YY^\dagger = \frac{1}{\kappa^2}(\Delta\mathbb{I} - \tilde{H}[0])(X_{+1} + X_{+2})(\Delta\mathbb{I} - \tilde{H}^\dagger[0]), \quad (89)$$

$$ZZ^\dagger = \frac{1}{\kappa^2}(\Delta\mathbb{I} - \tilde{H}[0])X_{-2}(\Delta\mathbb{I} - \tilde{H}^\dagger[0]). \quad (90)$$

It is easy to verify that Eq. (82) is satisfied.

We note that if $\left(\chi\left(\frac{\tilde{H} - \tilde{H}^\dagger + i\tilde{\kappa}}{2i}\right)\chi^\dagger\right)_{11} = 0$, we modify the matrix in Eq. (72) as $h \to h + \rho\delta_{i1}\delta_{j1}$. The baths can be constructed by the above procedure with a small $\rho \to 0$.

## Data availability
The numerical data generated in this work is available from the authors upon reasonable request.

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

## Acknowledgments

We thank Liang Jiang, Douglas Stone, and Mengzhen Zhang for useful conversations. This work was supported by a grant from the Simons Foundation (Award number 505450, A.C.), and by the AFOSR MURI FA9550-15-1-0029.

## Author contributions

H.-K. L. and A. A. C. contributed equally in the development of idea, derivation of results, and writing of manuscript.

## Additional information

**Competing interests:** The authors declare no competing interests.

