## [Peer Review File · Nature Communications]

Reviewers' comments:

Reviewer #1 (Remarks to the Author):

This paper is mainly about quantum sensing with non-Hermitian Hamiltonian. The authors want to estimate a parameter in the Hermitian part of the Hamiltonian, and use a single-tone drive to control the system. The authors give the general expression for the signal-to-noise ratio (SNR), and applied the results to two-mode sensors. The authors find that the non-reciprocity can be a useful resource to enhance non-Hermitian sensing, instead of exceptional points.

I think the results are interesting, but there are several issues that need be clarified.

It seems that the authors focus only on estimating parameters in the Hermitian part of the Hamiltonian. As the Hamiltonian is non-Hermitian here, I feel that estimating the parameters in the anti-Hermitian part should be of more interest.

It seems to me that the authors mainly attempted to optimize the bath and the system-bath coupling to minimize the noise, due to different baths and system-bath couplings may lead to the same non-Hermitian Hamiltonian. My concern here is the justifiability of fixing the non-Hermitian Hamiltonian. Physically, one should first have the configuration of the system and bath, then study its dynamics. If one can change the bath and its coupling to the system, then why should the non-Hermitian Hamiltonian be constrained to be fixed?

From the view of metrology, I think the most important point is to improve the SNR or Fisher information. But it seems that the paper is mostly devoted to bounding the signal power S or the measurement time Γ , particularly for the two-mode sensor case. So I think it should be made clearer how it is related to enhancing the sensitivity of sensing.

On Pg. 5, the paper claims that there is no advantage of using multi-tone drive for the small epsilon region as the information generated by each tone is independent. I am a bit confused by this statement, as in the information theory, the total Fisher information is the sum of Fisher information of independent events, so it seems that the information from each tune can contribute to the total information about epsilon. Even if there is no correlation, one still gained more information from independent measurements.

For Eq. (33), the authors claimed that the signal power can be infinitely amplified by increasing χ_{12} while keeping the intra-cavity photon number fixed. I think this is quite interesting. But can the authors explain more about the physics mechanism why this can happen? And is there any limit on χ_{12} ?

In summary, I find this is an interesting paper, and it can possibly be published in Nat. Comm. But the above issues need be addressed first.

Reviewer #2 (Remarks to the Author):

In the manuscript entitled "Non-Hermitian Quantum Sensing: Fundamental Limits and Non-Reciprocal Approaches," the authors evaluate a recent scheme for achieving improved sensitivity scaling in exceptional point-based sensors (namely, square root dependence for second order EPs, for example). The original claim was that if very small perturbations cause resonance frequencies to shift with the square root of the perturbation (near the exceptional point), then this non-analytic behavior would lead to very high sensitivity. However, the authors of this manuscript point out that if two essential ingredients are incorporated (which have been previously neglected), namely (1) changes in the eigenvectors by the perturbation, and (2) noise effects, then exceptional points need not be the key ingredient to enhance sensitivity. In fact, the authors point out that generic

amplification does the trick. They derive corresponding bounds for the signal power and the measurement rate. They go on to show that if the system can be made non-reciprocal, then these bounds can be lifted and sensitivity can be significantly enhanced beyond any reciprocal realization. This is a highly timely work, which presents an important result that sheds light on (and calls into question) a set of major recent results on exceptional point-based sensing. The discussion on non-reciprocal devices proposes a new route towards enhanced sensing that is both original and convincingly presented. This is an excellent work, highly deserving of publication in Nature Communications.

Issues that I recommend the authors address:

- As the authors point out, they focus on the regime where the perturbation is smaller than the linewidth of the mode. They should be explicit about how their result changes in the opposite limit, namely where the coupled-cavity system is very low loss and weakly coupled to its input waveguide (i.e., towards the Hermitian case).
- As an example of a non-reciprocal system, the authors study the model of Eq. (35), a reasonable case. This model can be seen as fundamentally non-Hermitian because of the asymmetry in off-diagonal coupling. What about the case of non-reciprocal models whose only non-Hermiticity arises from standard amplification (i.e., non-Hermiticity on the diagonals)? Specifically, this would be the case where the coupling between resonators is complex (more than two resonators may be necessary for non-trivial behavior due to the possibility to gauge away the coupling phase). Since this type of non-reciprocity is quite different from that of Eq. (35), and since it has been realized in a range of experiments (particularly in driven systems), models like this should be commented upon in relation to the derived bounds.
- Appendix A is quite brief, and mostly just refers to the literature for the derivation of the non-Hermitian Hamiltonian. I would recommend either expanding it to include a somewhat full derivation, or removing it and just referring to the relevant textbooks (e.g., Ref. 23) in the main text.

Reply to referee 1

- It seems that the authors focus only on estimating parameters in the Hermitian part of the Hamiltonian. As the Hamiltonian is non-Hermitian here, I feel that estimating the parameters in the anti-Hermitian part should be of more interest.

We thank the referee for their suggestion, and for making us realize that our work is even more general than stated in the original manuscript. The general theory derived in our manuscript is indeed also directly applicable to a non-Hermitian perturbation in almost all cases of interest: one simply replaces V in Eqs. 23 and 25 by the non-Hermitian matrix V describing the generalized perturbation. The only situation where this would not be valid is the pathological case where the noise associated with this non-Hermitian perturbation does not vanish in the limit $\epsilon \rightarrow 0$, i.e. it remains finite if there is no perturbation. As long as this is not the case, any extra noise associated with the non-Hermitian perturbation will not impact our linear-response expressions (i.e. it would not change the expression for the signal to noise ratio in Eq. 25 to leading order in ϵ). **We have added a paragraph after Eq. (25) to highlight this fact.**

- It seems to me that the authors mainly attempted to optimize the bath and the system-bath coupling to minimize the noise, due to different baths and system-bath couplings may lead to the same non-Hermitian Hamiltonian. My concern here is the **justifiability of fixing the non-Hermitian Hamiltonian**. Physically, one should first have the configuration of the system and bath, then study its dynamics. If one can change the bath and its coupling to the system, then why should the non-Hermitian Hamiltonian be constrained to be fixed?

The referee seems to have perhaps slightly misinterpreted our overall intent here; we apologize if we were not sufficiently clear. The motivation for our work comes from recent studies of non-Hermitian Hamiltonians for sensing which completely ignore the role of fluctuations. One of our key goals was to show that for a given non-Hermitian Hamiltonian, quantum mechanics requires there to be minimal (non-zero) amount of noise, noise that will ultimately limit the SNR of our sensor. Our manuscript derives what these minimal amounts of noise are (given a particular non-Hermitian Hamiltonian), as this determines the best possible performance of any quantum sensor using the given non-Hermitian dynamics. We then use these results to investigate whether certain kinds of non-Hermitian Hamiltonians (e.g. those with exceptional points) are truly better sensors in the quantum regime, even in the best-case scenario. We believe the resulting bounds are useful and interesting, as they constrain the best one could possibly do given a certain linear dynamics. These bounds directly let us understand that there are fundamental constraints on any reciprocal sensor, bounds that can be surpassed by sensors that are non-reciprocal.

We of course agree that in a particular experiment, one cannot typically keep the non-Hermitian Hamiltonian fixed while at the same time tuning system-bath coupling

parameters; we never meant to suggest this. **To avoid confusion, we have added a sentence just after Eq. 25 to explain the above points.**

- From the view of metrology, I think the most important point is to improve the SNR or Fisher information. But it seems that the paper is mostly devoted to bounding the signal power S or the measurement time Γ , particularly for the two-mode sensor case. So I think it should be made clearer how it is related to enhancing the sensitivity of sensing.

We draw the referee's attention to Eq. (16), which we believe clearly shows the relation between the measurement rate Γ_{meas} and the SNR of our sensor: the measurement determines the rate with which the SNR grows in time. This meaning and importance of this equation is discussed at length in the paragraph immediately following Eq. 16. We also discuss (in the paragraph immediately after Eq. 16) the fundamental relationship between the SNR (and hence measurement) and the quantum Fisher Information, the standard metric for quantifying parameter sensing. We believe that these equations and discussion make it clear how the measurement rate is related to the sensing ability of our sensor.

- On Pg. 5, the paper claims that there is no advantage of using multi-tone drive for the small ϵ region as the information generated by each tone is independent. I am a bit confused by this statement, as in the information theory, the total Fisher information is the sum of Fisher information of independent events, so it seems that the information from each tune can contribute to the total information about ϵ . Even if there is no correlation, one still gained more information from independent measurements.

The referee is of course correct, but seems to miss a crucial aspect of our discussion here: we are interested in a situation where the total number of intra-cavity photons used for the measurement is constrained. When the system is driven by multiple drive tones, each tone independently contributes to the total cavity photon number. This is explicitly written in Eq. 22. The question then is how to "spend" these photons: is better to have them all be at one well chosen frequency, or is it advantageous to spread them over many frequencies, in order to map out a transmission curve. We show that it is always better to "spend" all of the photon budget on the frequency tone that gives the largest information, instead of spreading it among different tones. Note that if one did not constrain the total photon number, it would be impossible to make sensible comparisons, as one could always improve the SNR of any scheme indefinitely simply by increasing the drive power. **To avoid any misunderstanding, we have modified the paragraph in p.5 top left corner to highlight the importance and significance of constraining total photon number.**

- For Eq. (33), the authors claimed that the signal power can be infinitely amplified by increasing χ_{12} while keeping the intra-cavity photon number fixed. I think this is quite interesting. But can the authors explain more about the physics mechanism why this

can happen? And is there any limit on χ_{12} ?

We thank the referee for pointing out that we did not provide sufficient physical intuition for this result. Eq. (36) indicates that for our non-reciprocal two-mode sensor, χ_{12} can be made arbitrarily large by increasing the non-reciprocal coupling J . There is no fundamental limit on the size of this coupling (or χ_{12}), though of course in any experimental situation constraints stemming from details of the particular realization will set a limit. For example, if the non-reciprocal tunnelling is realized using a driven optomechanical system (as in the experiment of Ref. 38 in our work), limitations on the amount of auxiliary drive power that the system can handle will ultimately limit the maximum size of J .

As for the physical intuition, **we have added an entirely new schematic figure, Fig. 4, to help illustrate the origin of the effect.** The basic idea is that the parameter ϵ of interest “breaks” the directionality of our sensor system. For $\epsilon = 0$, the system is fully directional. Drive photons incident on cavity 1 are unable to reach cavity 2, and hence never experience the large coupling J which moves photons from mode 2 to mode 1. For $\epsilon \neq 0$, there is thus no amplification of the intracavity photon number as we increase J .

Now, if ϵ is made non-zero, photons incident on cavity one can tunnel from cavity one to 2. They can then return to cavity one via the large (amplifying) non-reciprocal tunnelling J . The largeness of J gives us a very strong response here to a non-zero ϵ . We thus have the central result: the non-reciprocity of the system and largeness of J means that the effects of our perturbation (i.e. non-zero ϵ) are strongly amplified, whereas the intracavity photon number is not amplified. **The caption of the newly-added Fig. 4 goes through the above explanation.**

Reply to referee 2

- As the authors point out, they focus on the regime where the perturbation is smaller than the linewidth of the mode. They should be explicit about how their result changes in the opposite limit, namely where the coupled-cavity system is very low loss and weakly coupled to its input waveguide (i.e., towards the Hermitian case).

The referee seems to be asking here about the regime where the size of the perturbation to be sensed is large, in particular larger than the coupling rate to the main input-output waveguide used to drive and probe the system. As discussed in our manuscript, our analysis is mostly focused on the standard regime of detecting an infinitesimal parameter change, where standard tools like linear response and perturbation theory are valid, and where standard metrics like the Fisher information can be applied. The validity of this approach (i.e. how small must ϵ be) depends on the details of the system, but in many cases one does indeed require that ϵ is smaller than the coupling rate κ to the input-output waveguide.

Going beyond a perturbative, linear response regime would of course be interesting; however, this regime tends to be more system specific, and deriving general bounds can be challenging. Given the ubiquity and utility of the weak-coupling linear response regime, we prefer to leave the nonlinear analysis to a future work.

Note however that we do comment at least partially on nonlinear sensing regimes with a large epsilon. In the section “Non-reciprocal sensors and the mode-splitting technique” (starting on p. 9), we discuss a non-reciprocal sensor in the regime where epsilon is not small compared to kappa. Specifically, we show that non-reciprocity can enhance the perturbation-induced splitting of mode frequencies in this regime. We stress that the vast majority of existing works on non-Hermitian sensing also focus on the regime of a small perturbation, as the square-root scaling that is employed only holds for a small epsilon.

- As an example of a non-reciprocal system, the authors study the model of Eq. (35), a reasonable case. This model can be seen as fundamentally non-Hermitian because of the asymmetry in off-diagonal coupling. What about the case of non-reciprocal models whose only non-Hermiticity arises from standard amplification (i.e., non-Hermiticity on the diagonals)? Specifically, this would be the case where the coupling between resonators is complex (more than two resonators may be necessary for non-trivial behavior due to the possibility to gauge away the coupling phase). Since this type of non-reciprocity is quite different from that of Eq. (35), and since it has been realized in a range of experiments (particularly in driven systems), models like this should be commented upon in relation to the derived bounds.

The referee is correct in anticipating that non-reciprocity needed for enhanced sensing could be obtained in a system with more than two mode, where all the non-Hermiticity is “diagonal” (i.e. local damping and/or anti-damping of each mode), and where the non-reciprocity is just due to having phases in the couplings between modes. The prescription for how such a three mode system can effectively realize a two-mode system with non-reciprocal tunnelling (like in Eq. 35) is discussed at great length in Ref. 29 (which is cited below Eq. 35). One starts with three modes (the two main modes and an auxiliary mode) in a ring geometry, with tunneling phases that enclose a synthetic flux corresponding to an effective Aharonov-Bohm phase of $\pi/2$. By tuning the amplitudes of the tunneling, and taking the limit of large damping of the auxiliary mode in the ring, one can obtain an effective two-mode Hamiltonian of the form given in Eq. 35. This recipe has even been implemented directly in, e.g., quantum optomechanics, see Ref. 38.

The referee’s comment here makes us realize that more details on this possible realization of Eq. 35 would be welcome by many readers. **We have thus added text below Eq. 35 indicating that the non-Hermitian, non-reciprocal tunnelling in this Hamiltonian can be realized through the simple combination of having Hermitian tunnelling phases in a three mode system, combined with damping and anti-damping of each mode.**

- Appendix A is quite brief, and mostly just refers to the literature for the derivation of the non-Hermitian Hamiltonian. I would recommend either expanding it to include a somewhat full derivation, or removing it and just referring to the relevant textbooks (e.g., Ref. 23) in the main text.

We prefer to keep the discussion in Appendix A in our paper for two important reasons. First, as many researchers interested in non-Hermitian sensing are not familiar with the quantum input-output theory, to make our paper self-contained, we think it important to have some details on the formalism we use. Second, even for readers familiar with this formalism, this Appendix removes any uncertainty or ambiguity about the parameters we use to characterize the system bath Hamiltonian. Although the derivation of Langevin equation is standard as discussed in e.g. Ref. [23], the definitions of some parameters may vary as authors and could often differ by a factor of 2, π , or complex phase. We believe readers would find this section helpful when they attempt to reproduce the result in our work. **We have changed the title of this section to reflect our purpose, and have also added text to make the derivation easier to follow.**

REVIEWERS' COMMENTS:

Reviewer #1 (Remarks to the Author):

The authors have satisfactorily answered all of my questions and concerns. I therefore recommend publication.

Reviewer #2 (Remarks to the Author):

I thank the authors for their response. I continue to recommend publication.